# SUPERCORRECT: ADVANCING SMALL LLM REASONING WITH THOUGHT TEMPLATE DISTILLATION AND SELF-CORRECTION

**Ling Yang**[1*✉], **Zhaochen Yu**[1*], **Tianjun Zhang**[4], **Minkai Xu**[5], **Joseph E. Gonzalez**[4]
**Bin Cui**[1†], **Shuicheng Yan**[2,3†]
[1]Peking University, [2]Skywork AI, [3]National University of Singapore,
[4]UC Berkeley, [5]Stanford University
Project: https://github.com/YangLing0818/SuperCorrect-llm

## ABSTRACT

Large language models (LLMs) like GPT-4, DeepSeek-R1, and ReasonFlux have shown significant improvements in various reasoning tasks. However, smaller LLMs still struggle with complex mathematical reasoning because they fail to effectively identify and correct reasoning errors. Recent reflection-based methods aim to address these issues by enabling self-reflection and self-correction, but they still face challenges in independently detecting errors in their reasoning steps. To overcome these limitations, we propose SUPERCORRECT, a novel two-stage framework that uses a large teacher model to *supervise* and *correct* both the reasoning and reflection processes of a smaller student model. In the first stage, we extract hierarchical high-level and detailed thought templates from the teacher model to guide the student model in eliciting more fine-grained reasoning thoughts. In the second stage, we introduce cross-model collaborative direct preference optimization (DPO) to enhance the self-correction abilities of the student model by following the teacher's correction traces during training. This cross-model DPO approach teaches the student model to effectively locate and resolve erroneous thoughts with error-driven insights from the teacher model, breaking the bottleneck of its thoughts and acquiring new skills and knowledge to tackle challenging problems. Extensive experiments consistently demonstrate our superiority over previous methods. Notably, our SUPERCORRECT-7B model significantly **surpasses powerful DeepSeekMath-7B by 7.8%/5.3% and Qwen2.5-Math-7B by 15.1%/6.3%** on MATH/GSM8K benchmarks, achieving new SOTA performance among all 7B models.

## 1 INTRODUCTION

Large language models (LLMs) (Brown et al., 2020; Anil et al., 2023; Achiam et al., 2023; Du et al., 2022; Jiang et al., 2024; Touvron et al., 2023a;b), such as GPT-4 (Achiam et al., 2023), DeepSeek-R1 (Guo et al., 2025), and ReasonFlux (Yang et al., 2025), have demonstrated significant improvements in various reasoning tasks. However, despite being pre-trained on large-scale mathematical datasets using diverse techniques, smaller models like Llama-3-8B (Dubey et al., 2024) and Qwen2.5-Math-7B (Yang et al., 2024a) continue to struggle with complex mathematical reasoning tasks.

Existing works aim to enhance the mathematical performance of LLMs through various approaches. We categorize these methods into two types: **traditional fine-tuning optimization** and **reflection-based optimization**. Traditional fine-tuning methods mainly focus on the exploration in training techniques like Supervised Fine-Tuning (SFT) (Roziere et al., 2023; Shao et al., 2024; Dubey et al., 2024), and LLM-alignment strategies like Reinforcement Learning from Human Feedback (RLHF) (Achiam et al., 2023; Ouyang et al., 2022; Bai et al., 2022a;b) and alternative methods like Direct Preference Optimization (DPO) (Rafailov et al., 2024). Although these methods have shown remarkable progress across a wide range of language tasks, their optimization objectives only focus on direct answers or simple reasoning rationales. Consequently, they struggle to locate the errors in the reasoning process and fail to revise the flawed reasoning logic of language models.

---

[*]Equal Contribution. ✉ yangling0818@163.com
[†]Corresponding authors.

Recent reflection-based methods attempt to address the shortcomings of fine-tuning methods and leverage the pre-designed prompts or general rules to instruct language models for self-reflection and self-correction during reasoning process (Shinn et al., 2024; Kim et al., 2024). Some methods (Li et al., 2023; 2024c) further employ LLMs to synthesize rule-based datasets for enhancing their self-correction abilities in training stage. However, as mentioned in Tyen et al. (2024), LLMs still struggle to independently identify errors in their reasoning steps. Without accurate error identifications, self-correction becomes more challenging. In complex mathematical reasoning, even when mistake locations are provided, LLMs often remain biased or misled by their previous reasoning context. Thus it remains difficult for language models to clarify the causes of reasoning errors within a single LLM.

To address these limitations, we propose a novel two-stage framework, namely **SUPERCORRECT**, utilizing a large teacher model's thoughts to *supervise* and *correct* both the reasoning and reflection processes of a smaller student model. As depicted in Figure 1, in the first stage, we extract *hierarchical thought template* from the teacher LLM to guide the student model in generating more fine-grained reasoning thoughts. The template contains a high-level thought providing a summarized and generalized solution for similar problems, and a detailed solution offering a detailed explanation of the critical reasoning steps. Compare to previous thought format such as CoT (Wei et al., 2022) and BoT (Yang et al., 2024b; 2025), our hierarchical thought templates offer deeper and more informative reasoning insights for later error corrections. In second stage, we propose *cross-model collaborative DPO* to optimize the student model and enhance its self-correction abilities by following the teacher's cross-model correction traces during training. Specifically, instead of merely simulating correct answers or preferred reasoning process, we instruct teacher LLM to identify and correct the error parts in student's thoughts. This cross-model correction trace is then used to guide the student model in performing better self-correction, enabling it to avoid and rectify specific errors. The critical insight of our cross-model DPO approach is enabling student language models to break the bottleneck of its thoughts and acquiring new error-driven insights and knowledge from teacher's correction traces.

Furthermore, we construct a high-quality fine-tuning dataset equipped with designed hierarchical thought templates containing 100k samples, and a pair-wise preference dataset for thought-level correction optimization containing 10k samples, which consists of: 1) a math problem, 2) prior reasoning steps in our pre-designed format, 3) the step with chosen analysis and corrective guidance, generated by teacher LLMs based on the ground truth solution 4) the step with rejected analysis and correction guidance, generated by student LLMs without access to the ground truth solution.

We summarize our contribution as follows: **(i)** We propose a novel two-stage fine-tuning method SUPERCORRECT for improving both reasoning accuracy and self-correction ability for LLMs. **(ii)** We propose hierarchical thought based fine-tuning to enable small-sized LLMs to produce more accurate and fine-grained reasoning thoughts. **(iii)** We propose cross-model collaborative DPO, which innovatively leverage SOTA LLMs to locate and correct the specific error thoughts in the reasoning process of smaller student LLMs, thus advancing their self-correction ability and breaking their thought bottleneck. **(iv)** We construct two high-quality datasets and develop three powerful reasoning LLMs SUPERCORRECT-Qwen/DeepSeek/Llama-7B, **achieving 70.2% accuracy on the MATH dataset and 89.5% on the GSM8K dataset, setting new SOTA performance among all 7B models**.

## 2  RELATED WORK

**Reinforcement Learning from Human Feedback for Large Language Models**   To improve the performance and reliability of LLMs, RLHF methods like Christiano et al. (2017) and Ouyang et al. (2022) are introduced for LLM alignment. This method is more demanding in dataset because it requires pairwise annotated data to train a reward model thus reflecting human preferences. And then train the policy model using reinforcement learning to maximize the estimated reward. Although this method proves to be effective, due to its reliance on the quality of reward model, this process is complex and computationally intensive. To simplify this process, Direct Preference Optimization (DPO) (Rafailov et al., 2024) was proposed which directly uses pair-wise data for optimization. By defining the preference loss as a function of the policy, DPO can optimize the policy using straightforward training techniques, avoiding the complexities of reinforcement learning. However, current methods only show limited improvements in mathematical reasoning due to the design of optimization unit. Works like Step-DPO(Lai et al., 2024) establish a more fine-grained reward unit by considering each intermediate reasoning step as a basic unit. However, they fail to clarify error causes and provide explicit guidance for correcting errors. In this paper, we specifically design a cross-model teacher-student collaborative thought-based reward, which takes each correction step as a basic optimization unit.

Figure 1: Overview of our proposed two-stage framework SUPERCORRECT. In the first stage, we extract hierarchical thought template from teacher LLM to supervise student LLM for producing more specific thoughts. In the second stage, we collect a dataset of paired self- and cross-correction traces for cross-model collaborative DPO.

**Reasoning with Self-Correction/Reflection** Self-correction for reasoning has shown promise in improving LLM outputs in terms of style and quality. Previous works (Li et al., 2023; Shinn et al., 2024; Madaan et al., 2024; Saunders et al., 2022; Miao et al., 2023; Chen et al., 2023a) focus on the concept of self-correction, i.e. having an LLM correct its own outputs. However, as mentioned in Huang et al. (2023), while self-correction may prove effective for improving model outputs in terms of style and quality, when it comes to reasoning tasks, LLMs struggle to identify and fix errors without external feedback. For example, Reflexion (Shinn et al., 2024) and RCI (Kim et al., 2024) both use ground truth correctness as a signal to halt the self-correction loop. Moreover, some attempts to self-correct logical or reasoning errors can sometimes turn correct answers into incorrect ones, resulting in worse overall performances (Huang et al., 2023). While previous works typically present self-correction as a process conducted within a specific LLM, our method leverage large-sized LLMs to explicitly identify the errors and gain correction insights from the errors. With this corss-model reward, we can revise the weaknesses exposed by small-sized LLMs during reasoning tasks through fine-tuning and correction-based preference optimization.

**Thought Expansion for Mathematical Reasoning** Thought expansion for reasoning mainly focus on pre-designed reasoning structure or template, which leverage prompting techniques to enhance mathematical reasoning capabilities of LLMs. Chain-of-Thought (CoT) prompting (Wei et al., 2022) and its variants (Kojima et al., 2022; Press et al., 2023; Arora et al., 2022), such as Least-to-Most (Zhou et al., 2022), Decomposed Prompting (Khot et al., 2022), and Auto-CoT (Zhang et al., 2022)—prompt LLMs to break down complex questions into simpler subtasks and systematically solve them before summarizing a final answer. Innovations like Tree-of-Thought (Yao et al., 2024) and Graph-of-Thought (Besta et al., 2024), have further complex this field by exploring dynamic, non-linear reasoning pathways to expand heuristic capabilities of LLMs (Chen et al., 2023b; Ning et al., 2023). Other methods like PoT (Chen et al., 2022), PAL (Gao et al., 2023b) and (Gou et al., 2023) attempt to utilize external tools such as code to avoid hallucination of LLMs in the mathematical reasoning process. However, they suffer from increased resource demands and greater time complexity, depend on manual prompt crafting, and are often tailored to specific task types. Recent BoT (Yang et al., 2024b) propose a task-agnostic paradigm with meta buffer to efficiently solve the problems based on accumulated thought templates. However, it is a training-free framework which may not essentially boost the reasoning ability of LLMs. To further improve the internal reasoning ability of LLMs, Quiet-STaR (Zelikman et al., 2024) uses RLHF-based self-teaching with LLMs' self-generated thoughts to improve reasoning in normal tasks and simple math problems. For more complex problems that are beyond the students' capabilities, this think-before-reasoning pattern may not work well. In this paper, we utilize a new cross-model paradigm to enable LLMs to boost both reasoning and self-correction abilities from external model feedbacks, thereby breaking the bottleneck of original thoughts of LLMs and broadening the model's capability to address a wider range of issues.

## 3 PRELIMINARY

**Reinforcement Learning from Human Feedback** Reinforcement Learning from Human Feedback (RLHF) (Christiano et al., 2017) is an effective approach for enhancing the robustness, factuality, and safety of LLMs (Ouyang et al., 2022). RLHF consists of three training phases: 1) supervised fine-tuning

(SFT); 2) reward model training, and 3) policy model fine-tuning. **SFT Phase**: RLHF typically begins by fine-tuning a pre-trained LM with supervised learning on high-quality data for the downstream task(s) of interest (dialogue, summarization, etc.), to obtain a model $\pi_{sft}$. **Reward Modelling Phase**: given any text, the reward model will assign a scalar reward value to the last token, and the larger the reward value, the better the sample. Following Stiennon et al. (2020), training reward models often involves utilizing a dataset comprised of paired comparisons between two responses generated for the same input. The modeling loss for each pair of preferred and dis-preferred samples is:

$$\mathcal{L}(\psi) = \log \sigma(r(x, y^+) - r(x, y^-)), \tag{1}$$

where $\sigma$ is the sigmoid function. $r$ represents the reward model with parameters $\psi$, and $r(x, y)$ is the a single scalar predicted reward for input prompt $x$ and response $y$. However, this method is often considered complex due to the complex training pipeline. **RL Fine-Tuning Phase**: During the RL phase, the learned reward function is used to provide feedback to the language model. Following prior works (Tutor; Jaques et al., 2020), the optimization is formulated as

$$\max_{\pi_\theta} \mathbb{E}_{x \sim \mathcal{D}, y \sim \pi_\theta(y|x)} \big[ r_\phi(x, y) \big] - \beta \mathbb{D}_{\text{KL}} \big[ \pi_\theta(y \mid x) \mid\mid \pi_{ref}(y \mid x) \big], \tag{2}$$

where $\beta$ is a parameter controlling the deviation from the base reference policy $\pi_{ref}$, namely the initial SFT model $\pi_{sft}$. In practice, the language model policy $\pi_\theta$ is also initialized to $\pi_{sft}$. Due to the discrete nature of language generation, this objective is not differentiable and is typically optimized with reinforcement learning. The standard approach (Ziegler et al., 2019; Bai et al., 2022a; Ouyang et al., 2022) has been to construct the reward function as metioned in Equation (1), and maximize using PPO Schulman et al. (2017).

**Direct Preference Optimization (DPO)** As an competitive alternative for traditional RLHF method, DPO (Rafailov et al., 2024) was introduced to directly leverage pair-wise preference to optimize the policy model with an equivalent optimization objective. Specifically, given an input prompt $x$, and a preference data pair $(y^+, y^-)$, DPO aims to maximize the probability of the preferred output $y^+$ and minimize that of the undesirable output $y^-$. The optimization objective is formulated as:

$$\mathcal{L}_{DPO}(\theta) = -\mathbb{E}_{(x, y^+, y^-) \sim D} [\log \sigma(\beta \log \frac{\pi_\theta(y^+|x)}{\pi_{ref}(y^+|x)} - \beta \log \frac{\pi_\theta(y^-|x)}{\pi_{ref}(y^-|x)})], \tag{3}$$

where $D$ is the pair-wise preference dataset, $\sigma$ is the sigmoid function, $\pi_\theta(\cdot|x)$ is the policy model to be optimized, $\pi_{ref}(\cdot|x)$ is the reference model kept unchanged during training, and the hyperparameter $\beta$ controls the distance from the reference model.

## 4 METHOD

### 4.1 SUPERVISED FINE-TUNING WITH HIERARCHICAL THOUGHT TEMPLATE

**Constructing Hierarchical Thought Templates from Teacher LLMs** The traditional instruction-response datasets for training LLMs (Ouyang et al., 2022) mainly focus on the correctness of the response, leading LLMs to merely simulate the provided solution and the answer, while ignoring the importance of the intermediate reasoning thought. Recent work such as BoT (Yang et al., 2024b) utilizes a high-level reasoning guideline (thought template) to enable LLMs to efficiently solve similar problems in a training-free manner. However, for complex and diverse mathematical reasoning tasks, we find that using only a high-level thought template is insufficient, especially for small-sized LLMs. To empower small LLMs to tackle complex reasoning tasks, we specifically design a **hierarchical thought template** extracted from large teacher LLMs for transfer to small student LLMs. This new hierarchical thought template comprises both *a high-level thought* and *a detailed solution*. The former provides a summarized and generalized solution for similar problems, while the latter offers a detailed explanation of the critical reasoning steps.

Based on this hierarchical thought template, we can propose a new fine-tuning objective that aims to incorporate human-like hierarchical problem-solving thought structures into the model reasoning and explicitly produce hierarchical thought during reasoning process. We first collect a set $D = \{(x, \hat{y}, \hat{s})\}$ of mathematical problems $x$ with ground-truth answers $\hat{y}$ and solution $\hat{s}$. For each problem $x \in D$, we first utilize our pre-defined prompt denoted as $P_{tea}$, as shown in the below text box, to extract hierarchical thought templates from teacher LLMs (e.g., SOTA LLMs like o1-preview/o1-mini). For more details about our prompt, we present all of our prompts in Appendix A.

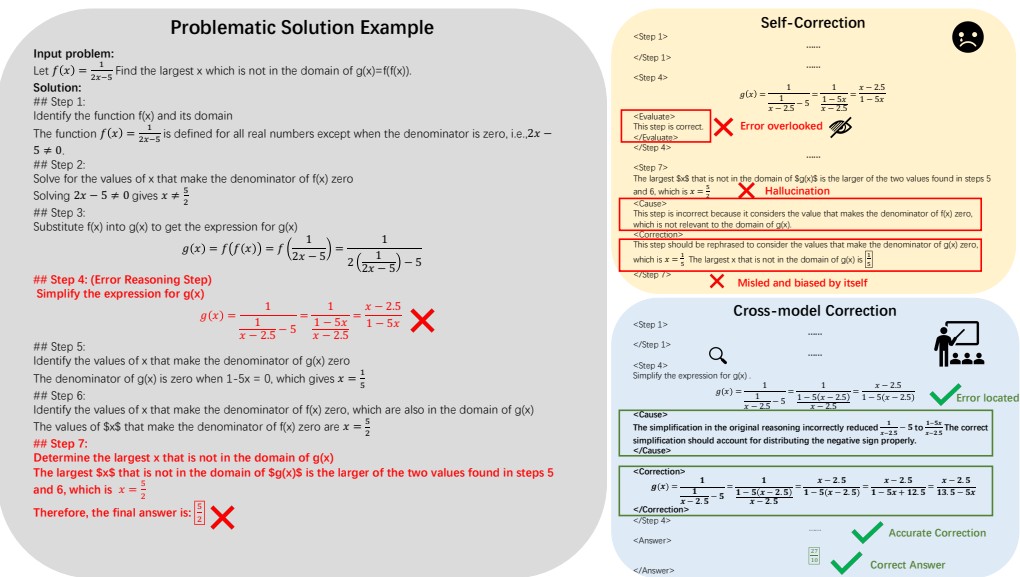

Figure 2: An illustrative comparison between self-correction and our cross-model correction. Cross-model correction can enable more precise error localization and thought correction.

> ***Prompt for Extracting Hierarchical Thought Template***
> *Transform the solution of the following math problem into a step-by-step XML format, each step should be enclosed within tags like $\langle Step1 \rangle \langle /Step1 \rangle$. For each step enclosed within the tags, determine if this step is challenging and tricky, if so, add detailed explanation and analysis enclosed within $\langle Key \rangle \langle /Key \rangle$ in this step, as helpful annotations to make the student better understand this step correctly thus mastering the solution. After all the reasoning steps, summarize the common solution and reasoning steps to help him generalize to similar problems within $\langle Generalized \rangle$ $\langle /Generalized \rangle$. Finally present the final answer enclosed within $\langle Answer \rangle \langle /Answer \rangle$.*

Then we can obtain the high-quality fine-tuning dataset $D_{sft}$ as:

$$D_{sft} = \pi_{tea}(P_{tea}, x, \hat{s}) = \{x, s_{tea}, T_{tea}, y_{tea} | x \in D\}, \tag{4}$$

where $s_{tea}$ is the formalized solution steps, $T_{tea}$ is the hierarchical thought for the solution, and $y_{tea}$ is the final answer extracted from $s_{tea}$. We provide an example of our hierarchical thought template in Appendix A. For normal and easy steps, we provide brief explanation and direct solution, as for tricky and difficult reasoning steps, we provide a **detailed solution** and in-depth explanation within $\langle Key \rangle$ which will help student LLMs to better grasp the insight within the detailed thought. Furthermore, we provide a **high-level thought** within $\langle Generalized \rangle$ as a generalized guidance which helps to efficiently solve similar problems.

**Thought-based Supervised Fine-tuning** After curating our thought-based dataset $D_{sft}$, our optimization objective is to make student LLMs $\pi$ reasoning with hierarchical thought and have a more comprehensive understanding for each problem-solving process, which can be formulated as:

$$\mathcal{L}_{\text{sft}} = \text{argmax} \sum_{(P_{stu}, x, T_{tea}, s_{tea}) \in D_{sft}} \log \pi((T_{tea}, s_{tea}) | (P_{stu}, x)). \tag{5}$$

Starting from the base student LLM $\pi$, $\mathcal{L}_{\text{sft}}$ maximizes the likelihood of response $(T_{tea}, s_{tea})$ given prompt $P_{stu}$ and input problem $x$, where $P_{stu}$ denotes the pre-defined prompt as $P_{tea}$. After the fine-tuning process, we greatly enhance the reasoning ability of base student LLMs by learning the hierarchical thought from SOTA reasoning LLMs and enable the student LLMs to produce similar hierarchical thought along with final answer. Then, we obtain fine-tuned student LLMs $\pi_{ref}$ that could be used for cross-model collaborative dpo in Section 4.2.

## 4.2 CROSS-MODEL COLLABORATIVE DPO

**Boosting DPO with Thought Correction**   While DPO proves to be effective in some areas (e.g., chat, style, etc.), its optimization objective is less effective for complex mathematical reasoning tasks. As noted in Lai et al. (2024), the issue arises because errors in solving complex mathematical problems often occur at the most challenging steps (e.g., complicated calculations, tricky transformations). This may lead to wrong optimization during training, as correct previous steps are also rejected. Furthermore, it is challenging for a single LLM to detect and correct its own errors (Tyen et al., 2024). This is akin to students struggling to gain insights from their own incorrect solutions. The root of the error lies in flawed reasoning, making it inefficient to merely imitate the correct solution without addressing the underlying thought-level mistakes. To address this, we have carefully designed novel and fine-grained optimization objectives that prioritize thought-level correction over traditional instance-level preference. Specifically, we first accurately locate the error step and then use the correction trace of this error step as the optimization unit. This approach prioritizes cross-model correction traces from teacher LLMs $\pi_{tea}$ over self-correction traces from student LLMs $\pi_{ref}$, thereby enhancing the error detection and self-correction abilities of student LLMs.

**Collecting Error Thoughts and Corrections**   To achieve thought-level correction, we need to collect a dataset containing fine-grained paired data of self- and cross-correction traces. Specifically, we utilize the fine-tuned student LLM $\pi_{ref}$ to conduct thought-based reasoning on our sampled test dataset denoted as $D_{test} = \{x_{test}, \hat{y}_{test}, \hat{s}_{test}\}$, and we obtain the test results denoted as $\pi_{sft}(x_{test}) = \{x_{test}, s_{test}, T_{test}, y_{test} | x_{test} \in D_{test}\}$. After filtering out erroneous problem-solution pairs that satisfy $y_{test} \neq y_{t\hat{e}st}$ and finally obtain the erroneous dataset:

$$D_{err} = \{x_{test}, \hat{y}_{test}, \hat{s}_{test}, s_{err}, T_{err}, y_{err} | x_{test} \in D_{test}\}, \tag{6}$$

here $s_{err}$ is the error solution and $T_{err}$ is the corresponding error thought, $y_{err}$ represents the error answer extracted from $s_{err}$. Given that each erroneous solution is explicitly presented as a sequence of reasoning steps $s_{err} = s_1, s_2, \ldots, s_n$, we proceed to verify the correctness of each reasoning step until we find the first error and record its step number $k$. Here we utilize current powerful models (e.g., gpt-4o, o1-mini) in mathematical reasoning to function as an experienced teacher model $\pi_{tea}$. To obtain the corresponding error steps and cause analysis, we design a prompt $P_c$ to instruct $\pi_{tea}$ to search for the logic flaws and errors in the provided reasoning steps. After searching $s_{err}$ and evaluating each reasoning steps, we could locate each error steps and annotate each error step with error cause analysis $a_i$ and correction guidance $c_i$. Thus we could obtain an annotated dataset of pair-wise self- and cross-corrections:

$$D_{corr} = \{(x, \{s_i\}_{i=0}^{k-1}, (a_k^+, c_k^+), (a_k^-, c_k^-),) | x \in D_{err}\}, \tag{7}$$

where $k$ denotes the first error step. Here $(a_k^+, c_k^+)$ is chosen as the corrected step with analysis from teacher model, $(a_k^-, c_k^-)$ is chosen as the rejected correction step and cause analysis from the student model, utilizing the same correction prompt as the teacher. To further ensure the quality of our dataset, we additionally propose an inspector LLM to conduct iterative evaluation which verifies the accuracy of the correction trace by comparing it against the input problem and the ground-truth solution. If issues are detected, the problematic parts are sent back to the teacher LLMs for revision. This iterative checking process continues until no errors remain, with a maximum of three iterations allowed. In our implementation, we apply inspector LLM both in the curation process of HSFT dataset and pair-wise self-and corrections dataset. For more detail, please refer to Appendix D, we also make detailed analysis of the dataset quality in Appendix D.2.

**Improving Self-correction Ability with Cross-model Correction**   In the second stage of our method, our proposed **cross-model collaborative DPO** leverages cross-model correction from teacher LLMs to enhance the error detection and self-correction ability of student LLMs. As noted in Equation (7), the previous $k-1$ correct reasoning steps $\{s_i\}_{i=0}^{k-1}$ are combined with input problem $x$, our cross-model collaborative DPO aims to maximize the probability of the teacher LLM's correction and analysis of the error step $(a_k^+, c_k^+)$, while minimizing the probability of the student LLM's self-correction and analysis $(a_k^-, c_k^-)$. The optimization objective of our cross-model collaborative DPO can be formulated as:

$$\mathcal{L}_{\text{Cross-DPO}}(\theta) =$$

$$-\mathbb{E}_{(x, s_{1 \sim k-1}, (a_k^+, c_k^+)) \sim D_{corr}} \left[ \log \sigma \left( \beta \log \frac{\pi_\theta((a_k^+, c_k^+) | x; s_{1 \sim k-1})}{\pi_{ref}((a_k^+, c_k^+) | x; s_{1 \sim k-1})} - \beta \log \frac{\pi_\theta((a_k^-, c_k^-) | x; s_{1 \sim k-1})}{\pi_{ref}((a_k^-, c_k^-) | x; s_{1 \sim k-1})} \right) \right].$$

$$\tag{8}$$

By prioritizing cross-model correction over self-correction, as illustrated in Figure 2, our method helps student model to accurately locate the erroneous steps of the mathematical reasoning process and effectively conduct self-correction. Furthermore, this process also helps the student LLMs to rectify its original flawed thoughts and avoid specific errors thus improving the reasoning ability and mitigate hallucination problems.

Table 1: Quantitative comparison. Models are evaluated with chain-of-thought reasoning using open-source evaluation framework (Gao et al., 2023a) [†]. "general" denotes whether the model is for general tasks or designed for specific tasks. "open" denotes open-source or not.

| Model | size | general | open | MATH (%) | GSM8K (%) |
|---|---|---|---|---|---|
| GPT-3.5-Turbo | - | ✓ | ✗ | 42.5 | 92.0 |
| Gemini-1.5-Pro (Reid et al., 2024) | - | ✓ | ✗ | 67.7 | 90.8 |
| Claude-3-Sonnet | - | ✓ | ✗ | 71.1 | 96.4 |
| GPT-4-1106 (Achiam et al., 2023) | - | ✓ | ✗ | 64.3 | 91.4 |
| GPT-4-Turbo-0409 (Achiam et al., 2023) | - | ✓ | ✗ | 73.4 | 93.7 |
| GPT-4o-0806 | - | ✓ | ✗ | 76.6 | 95.8 |
| Llama-3-8B-Instruct (Touvron et al., 2023a) | 8B | ✓ | ✓ | 30.0 | 79.6 |
| Qwen2-7B-Instruct (Yang et al., 2024a) | 7B | ✓ | ✓ | 49.6 | 82.3 |
| Llama-3-70B-Instruct (Touvron et al., 2023a) | 70B | ✓ | ✓ | 50.4 | 93.0 |
| DeepSeek-Coder-V2-Instruct (Zhu et al., 2024) | 236B | ✗ | ✓ | 75.7 | 94.9 |
| Code-Llama-7B (Roziere et al., 2023) | 7B | ✗ | ✓ | 13.0 | 25.2 |
| MAmooTH-CoT (Yue et al., 2023) | 7B | ✗ | ✓ | 10.4 | 50.5 |
| WizardMath (Luo et al., 2023) | 7B | ✗ | ✓ | 10.7 | 54.9 |
| MetaMath (Yu et al., 2023) | 7B | ✗ | ✓ | 19.8 | 66.5 |
| MetaMath-Mistral-7B (Yu et al., 2023) | 7B | ✗ | ✓ | 28.2 | 77.7 |
| MathScale-Mistral Tang et al. (2024) | 7B | ✗ | ✓ | 35.2 | 74.8 |
| InternLM-Math-7B (Ying et al., 2024) | 7B | ✗ | ✓ | 34.6 | 78.1 |
| Xwin-Math-Mistral-7B (Li et al., 2024a) | 7B | ✗ | ✓ | 43.7 | 89.2 |
| MAmmoTH2-7B-Plus (Yue et al., 2024) | 7B | ✗ | ✓ | 45.0 | 84.7 |
| MathGenieLM-Mistral (Lu et al., 2024) | 7B | ✗ | ✓ | 45.1 | 80.5 |
| InternLM-Math-20B (Ying et al., 2024) | 20B | ✗ | ✓ | 37.7 | 82.6 |
| MathGenieLM-InternLM2 (Lu et al., 2024) | 20B | ✗ | ✓ | 55.7 | 87.7 |
| Meta-Llama3.1-8B-Instruct (Dubey et al., 2024) | 8B | ✗ | ✓ | 51.9 | 84.5 |
| **SUPERCORRECT-Llama-8B (Ours)** | 8B | ✗ | ✓ | **58.2** | **89.7** |
| DeepSeekMath-7B-Instruct(Shao et al., 2024) | 7B | ✗ | ✓ | 46.8 | 82.9 |
| **SUPERCORRECT-DeepSeek-7B (Ours)** | 7B | ✗ | ✓ | **54.6** | **88.2** |
| Qwen2.5-Math-7B-Instruct (Yang et al., 2024a) | 7B | ✗ | ✓ | 55.1 | 83.2 |
| **SUPERCORRECT-Qwen-7B (Ours)** | 7B | ✗ | ✓ | **70.2** | **89.5** |

[†] lm-evaluation: https://github.com/EleutherAI/lm-evaluation-harness.

# 5 EXPERIMENTS

## 5.1 EXPERIMENTAL SETUP

**Base Models, Datasets and Evaluations** We apply SUPERCORRECT to different base models to demonstrate its generalization ability and achieve new SOTA results, including recent powerful **Qwen2.5-Math-7B** (Yang et al., 2024a), **Meta-Llama3.1-8B** (Dubey et al., 2024), **DeepSeek-Math-7B** (Liu et al., 2024), these models have been recognized to be reasoning-efficient with smaller size and strong reasoning ability especially in mathematical problems. In the SFT stage, we use mathematical problems from the training set of Math (Hendrycks et al., 2021) which consists of 7500 challenging competition mathematics problems, and training set of GSM8K (Cobbe et al., 2021) consists of 7473 high quality linguistically diverse grade school math word problems. Furthermore, we additionally translated 670 challenging math problems from GaoKao Bench (Zhang et al., 2023a) which is based on Chinese 2010-2022 GAOKAO examinations. To further enrich the diversity of our dataset, we sampled some

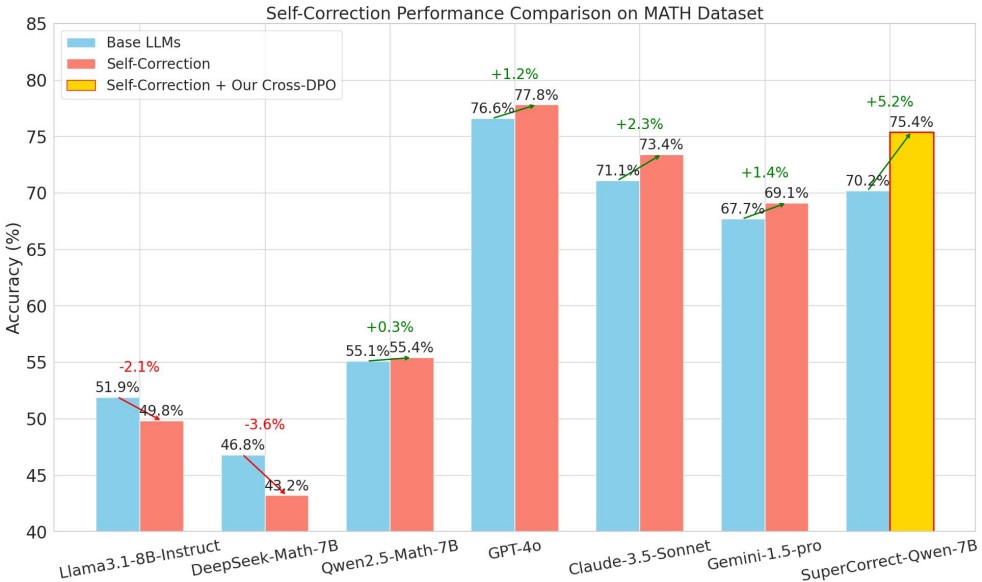

Figure 3: Comparison between different models and our SUPERCORRECT. Here we chose SUPERCOR-RECT-Qwen-7B as our model. The differences of the accuracy has been marked by arrows with different colors, red means accuracy decreased, and green means accuracy improved.

challenging problems from NuminaMath (Li et al., 2024b) and MetaMath(Yu et al., 2023). To align with our hierarchical thought reasoning process, we leverage SOTA LLMs o1-mini/gpt-4o-mini to create hierarchical thought based on the ground truth solution as mentioned in Section 4.1, and establish a hierarchical thought based dataset. In the Cross-model DPO stage, we collect 20k incorrect reasoning results from three different SFT models and processed as described in Section 4.2. For evaluation, we use the test set from **MATH** (Hendrycks et al., 2021) and **GSM8K** (Cobbe et al., 2021) datasets, and test chain-of-thought reasoning accuracy utilizing open-source evaluation framework (Gao et al., 2023a).

**Implementation Details**    We conduct our experiments on 8 NVIDIA A100-PCIE-40GB GPUs. Here we denote our hierarchical thought based supervised fine-tuning as HSFT for simplicity. Initially, we utilize the 100K HSFT data for hierarchical thought supervised fine-tuning on the base models to obtain our HSFT models. We train all of our models for 4 epochs, with training batch size set to 8 and gradient accumulation steps set to 16. The learning rate is set to $2e^5$ and we use AdamW optimizer along with the cosine learning rate scheduler. The warmup ratio is set to 0.02 and we use flash-attention (Dao et al., 2022) to save GPU memory. Subsequently, we perform Cross-model DPO based on the HSFT models. For Cross-model DPO, we train for 8 epochs, with a global batch size of 128 and a learning rate of $1 \times 10^{-6}$. And we use the AdamW optimizer along with cosine learning rate scheduler, and the warmup ratio is set to 0.05.

Table 2: Accuracy comparison between different methods, here we choose Qwen2.5-Math-Instruct as Base model denoted as Base and our Cross-model DPO is denoted as Cross-DPO. Here we separately compare our first HSFT stage with traditional SFT method and Cross-DPO stage with Reflexion(Shinn et al., 2024). We show the improved accuracy in green compare to previous methods. We provide quantitative results with more base LLMs (i.e., Llama3.1 and DeepSeek-Math) in Table 7 of Appendix E.

| Model | Base | Base + SFT | Base + HSFT | Base-HSFT + Reflexion | Base-HSFT + Cross-DPO |
|---|---|---|---|---|---|
| **MATH** (%) | 55.1 | 57.4 | **62.4** (+5.0) | 63.1 | **70.2** (+7.1) |
| **Model** | Base | Base + SFT | Base + HSFT | Base-HSFT + Reflexion | Base-HSFT + Cross-DPO |
| **GSM8K** (%) | 83.2 | 84.3 | **87.2** (+2.9) | 86.8 | **89.5** (+2.7) |

## 5.2 MAIN RESULTS

**Enhanced Reasoning Accuracy**    As shown in Table 1, our method **achieves new SOTA performance among all 7B models, significantly surpassing powerful DeepSeekMath-7B by 7.8% and Qwen2.5-Math-7B by 15.1% on MATH benchmark**. This promising results demonstrates our superiority and effectiveness in handling complicated reasoning tasks. Notably, we can achieve better results than larger-sized models such as Llama3-70B-Instruct (Touvron et al., 2023a) in GSM8K and MATH, and achieve accuracy comparable to GPT-4o and GPT-4o-mini with our best model SUPERCORRECT-Qwen-7B. We

Figure 4: Improvement comparison between different topics. Here we chose Qwen2.5-Math-7B-Instruct and our SUPERCORRECT-Qwen-7B to show the improvement in performance of different mathematical problem Types. The part in green is the improved part of our SUPERCORRECT, and the part in black is the original reasoning accuracy of Qwen2.5-Math-7B-Instruct.

attribute this improvement in reasoning accuracy in two folds: 1) The first HSFT stage that equips student LLMs with a deeper and fine-grained reasoning process. Compare to conventional CoT reasoning process, it helps the student LLMs to think more carefully thus improving the reasoning consistency and reduce hallucinations issues on the problems that the student LLMs already mastered. 2) The second cross-model DPO stage that leverages the error-driven insights from teacher LLM to help student LLMs break the bottleneck of their thoughts thus making it possible to deal with the problems that the student LLMs in acquiring the skills and knowledge to tackle problems they were previously unable to solve. We also present some detailed examples of hierarchical reasoning in Appendix F from different datasets, please check them to have a comprehensive understanding of our SUPERCORRECT.

**Improved Self-Correction Ability**   Here we also show the improved self-correction ability of our SUPERCORRECT as shown in Figure 3. After initial reasoning stage, we let all the LLMs to verify the reasoning process and detect the logic flaws and errors within each reasoning step, and try to correct them. As a result of self-correction, our SUPERCORRECT further increase the accuracy by 5∼6%, while other LLMs are ineffective to increase the accuracy, and some LLMs even decrease the original accuracy. Because our Cross-model DPO helps the LLMs to accurately locate the errors and logic flaws within each steps by learning teacher's correction traces, and use a fine-grained analysis and correction to help LLMs better correct them. After the Cross-model DPO process, the LLMs are not only able to consistently solve problems within its capabilities, but they are also able to solve wider range of problems with error-driven insights gained from teacher LLMs. We provide more quantitative analysis in Table 6 on how far cross-model DPO brings the student model and the teacher model closer to each other. We also provide some self-correction examples from different datasets, for more detail, please check Appendix G.

**Ablation Study**   We conduct ablation study of our SUPERCORRECT and put results in Table 2. As we can see, the improvement of traditional SFT is limited compare to our HSFT, which falls behind by 5% in accuracy. Based on our HSFT models, we further apply some self-correction methods such as Reflexion (Shinn et al., 2024) to compare with our Cross-DPO. From the results, we can find that our method wins again with lead of 7% in accuracy compare to Reflexion. These promising results demonstrate the effectiveness of our HSFT and cross-model DPO. Here we take an illustrative example in Table 3 of Appendix B for better understanding of our effective hierarchical thought reasoning. The CoT prompting method shows misunderstanding of "empty set" as it fails to account for the fact that the 512 sets already include the empty set. Equipped with our hierarchical thought-based reasoning (denoted as HT in Appendix A), we can see that the model realizes that the 512 sets include empty set. However, it fails to correctly recall the fact that the problem requires to include the empty set in the final

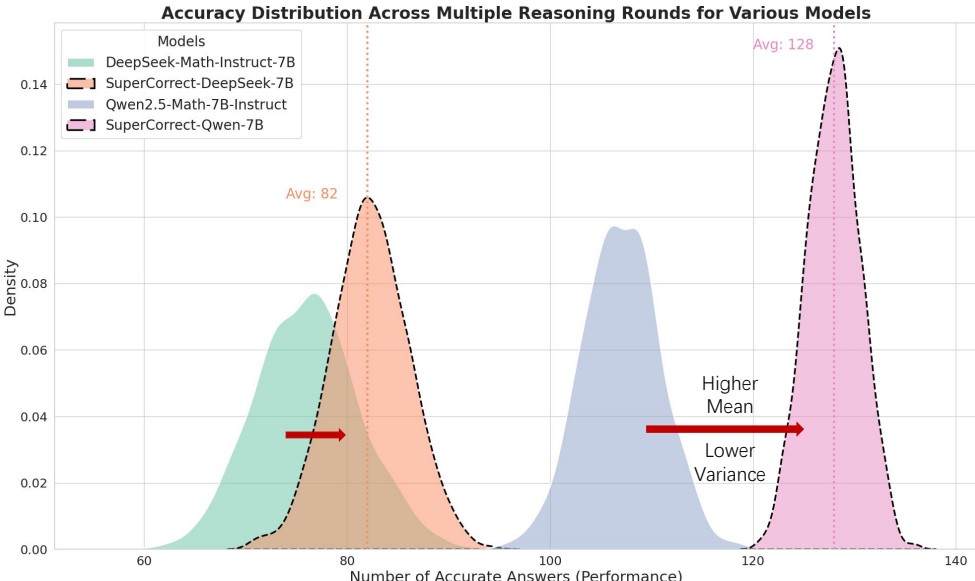

Figure 5: Quantitative analysis on reasoning stability. The higher mean value denotes higher average accuracy rate, and lower variance denotes higher reasoning stability.

answer, which is caused by hallucination issue. Finally, our HSFT LLMs could correctly resolve the problem with accurate understanding of empty set and avoid the hallucination issue.

**SupperCorrect Breaks Thought Bottleneck**  The problems within MATH dataset encompass a wide range of seven topics including algebra, counting & probability, intermediate algebra, number theory, geometry, prealgebra and precalculus. During our experiments, we observe that the accuracy for each topics are quiet different. For most LLMs, they tend to show better performance on algebra and prealgebra, but for other topics, it always show degradation in accuracy because they may have some thought bottleneck on those topics. As shown in Figure 4, our SUPERCORRECT improves the reasoning performance on all topics. It is noted that for the topics which are originally difficult for LLMs, it shows a more significant improvement compare to topics that the models are already mastered. This is because we utilize the error-driven insights during the Cross-model DPO stage to break the original thought bottleneck of LLMs, thus enlightening them with new techniques and tricks to solve the problems that they used have no idea to solve. The results further proves that our SUPERCORRECT could help to break the original thought bottleneck thus significantly improve the reasoning ability of LLMs, and narrowing the performance gap for different topics. More detail reasoning and self-correction results can be found in Appendix F. and Appendix G.

**SuperCorrect Achieves Better Reasoning Stability**  The test set of MATH dataset consists of 5000 problems in 5 different difficulty levels. To further evaluate the reasoning stability of our method, we additionally sample 300 problems of level-5 (hardest) from MATH test dataset. We conduct a quantitative analysis by repeating the experiment 256 times and compute the mean and variance of accuracy as shown in Figure 5. We can observe that, compare to the base model, our SUPERCORRECT helps to achieve higher mean value of accuracy rate. Moreover, our SUPERCORRECT significantly reduce the variance of accuracy distribution of multiple reasoning times. These phenomenons demonstrate our SUPERCORRECT can effectively improve both accuracy and stability for difficult reasoning problems.

## 6  CONCLUSION

In this paper, we propose SUPERCORRECT, a novel two-stage framework that significantly improve both reasoning and reflection processes of language models. In SUPERCORRECT, We propose hierarchical thought-based fine-tuning to enable LLMs to produce more fine-grained reasoning thoughts and introduce cross-model collaborative DPO to enhance the self-correction abilities of the student LLMS by following the teacher's correction traces. Extensive experiments consistently demonstrate our superiority over previous methods, surpasses powerful DeepSeekMath-7B by 5.3%∼7.8% and Qwen2.5-Math-7B by 6.3%∼15.1% on MATH and GSM8K benchmarks. For future work, we will generalize this new framework to larger models and more complex datasets.

ACKNOWLEDGEMENT

This work is supported by National Natural Science Foundation of China (U23B2048, U22B2037), Beijing Municipal Science and Technology Project (Z231100010323002), research grant No. SH2024JK29 and High-performance Computing Platform of Peking University and in part by NUS Start-up Grant A-0010106-00-00.

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

# A  ADDITIONAL PROMPTING DETAILS AND EXAMPLE

---

**Hierarchical Thought Template**

$\langle$Step 1$\rangle$

......

$\langle$/Step 1$\rangle$

......

$\langle$Step 4$\rangle$ **Calculate the Number of Ways to Roll Exactly 2 Sixes**
$\langle$Key$\rangle$
♠**Starting point of detailed solution**♠:
**Understanding Combinations and Independent Events**
The most challenging step is determining the number of ways to roll exactly two sixes. This involves two key concepts:

1. **Combinations** ($\binom{5}{2}$): This represents the number of ways to choose which two out of the five rolls will be sixes.

2. **Independent Choices for Remaining Rolls** ($5^3$): For the other three rolls that are not sixes, each has 5 possible outcomes (1 through 5).

By combining these, the total number of ways to get exactly two sixes is:

$$\binom{5}{2} \times 5^3$$

♠**End point of detailed solution**♠:
$\langle$/Key$\rangle$
$\langle$/Step 4$\rangle$
$\langle$Step 5$\rangle$

......

$\langle$/Step 5$\rangle$
$\langle$Step 6$\rangle$ **Calculate the Probability**
The probability of getting at most two sixes in five rolls is the ratio of the number of favorable outcomes to the total number of possible outcomes:

$$\frac{\binom{5}{0} \times 5^5 + \binom{5}{1} \times 5^4 + \binom{5}{2} \times 5^3}{6^5} = \frac{625}{648}$$

$\langle$/Step 6$\rangle$
$\langle$Generalized$\rangle$
◇ **Starting point of high-level thought**
**Generalizing to Similar Problems**
When dealing with probabilities of obtaining a certain number of specific outcomes in multiple trials (like rolling a die several times), follow these steps:

1. **Identify Total Outcomes**: Calculate the total number of possible outcomes using $n^k$.

2. **Determine Favorable Outcomes**: For each desired number of specific outcomes (e.g., exactly $r$ sixes), use combinations and multiply by the number of ways the remaining trials can occur.

3. **Sum Relevant Cases**: If the problem asks for "at most" or "at least," sum the favorable outcomes accordingly.

4. **Compute Probability**: Divide the total favorable outcomes by the total possible outcomes.

This approach is based on the **binomial probability formula**, which is widely applicable in scenarios with independent trials.
◇ **End point of high-level thought**
$\langle$/Generalized$\rangle$
$\langle$Answer$\rangle$ The probability of rolling a six in at most 2 of the 5 rolls is $\frac{625}{648}$. $\langle$/Answer$\rangle$

---

**Prompt for Extracting Hierarchical Thought Template**
Transform the solution of the following math problem into a step-by-step XML format, each step should be enclosed within tags like $\langle$Step1$\rangle\langle$/Step1$\rangle$. For each step enclosed within the tags, determine if this step is challenging and tricky, if so, add detailed explanation and analysis enclosed within $\langle$Key$\rangle\langle$/Key$\rangle$ in this step, as helpful annotations to make the student better understand this step correctly thus mastering the solution. After all the reasoning steps, summarize the common solution and reasoning steps to help him generalize to similar problems within $\langle$Generalized$\rangle\langle$/Generalized$\rangle$. Finally present the final answer enclosed within$\langle$Answer$\rangle\langle$/Answer$\rangle$.

- - - - - - - - - - - - - - - - - - - - - - - - - - - - - - - - - -

**Hierarchical Thought-based Reasoning Prompt (HT):**
Solve the following math problem in a step-by-step XML format, each step should be enclosed within tags like $\langle$Step1$\rangle\langle$/Step1$\rangle$. For each step enclosed within the tags, determine if this step is challenging and tricky, if so, add detailed explanation and analysis enclosed within$\langle$Key$\rangle\langle$/Key$\rangle$ in this step, as helpful annotations to help you thinking and remind yourself how to conduct reasoning correctly. After all the reasoning steps, summarize the common solution and reasoning steps to help you and your classmates who are not good at math generalize to similar problems within $\langle$Generalized$\rangle\langle$/Generalized$\rangle$. Finally present the final answer within $\langle$Answer$\rangle\langle$/Answer$\rangle$.

- - - - - - - - - - - - - - - - - - - - - - - - - - - - - - - - - -

**Grounded Correction Trace Prompt:**
First transform the Reasoning steps to be Checked into our required XML format as follow: for each step, the steps should be within corresponding tags like $\langle$Step1$\rangle\langle$/Step1$\rangle$, and next based on the problem and reference solution, evaluate each steps and find the fundamental logic flaws and errors in the given reasoning steps, if error detected, using $\langle$Cause$\rangle\langle$/Cause$\rangle$ to give a Refined and Concise explanation for the error cause within the corresponding Step tags along with $\langle$Correction$\rangle\langle$/Correction$\rangle$ to correct the error step and output correct step. And finally, present the correct final answer within $\langle$Answer$\rangle\langle$/Answer$\rangle$. Output All the transformed reasoning steps from $\langle$Step1$\rangle\langle$/Step1$\rangle$.

- - - - - - - - - - - - - - - - - - - - - - - - - - - - - - - - - -

**Correction Trace Prompt:**
First transform the Reasoning steps to be Checked into our required XML format as follow: for each step, the steps should be within corresponding tags like $\langle$Step1$\rangle\langle$/Step1$\rangle$, and next based on the problem, evaluate each steps and find the fundamental logic flaws and errors in the given reasoning steps, if error detected, using $\langle$Cause$\rangle\langle$/Cause$\rangle$ to give a Refined and Concise explanation for the error cause within the corresponding Step tags along with $\langle$Correction$\rangle\langle$/Correction$\rangle$ to correct the error step and output correct step. And finally, present the correct final answer within $\langle$Answer$\rangle\langle$/Answer$\rangle$. Output All the transformed reasoning steps from $\langle$Step1$\rangle\langle$/Step1$\rangle$.

As shown above, we present our meticulously designed prompt template used in our experiments along with an example of our hierarchical thought template. The prompt for extracting hierarchical thought template is designed for teacher LLMs to transform the original solution into hierarchical thought template. And for hierarchical thought-based reasoning prompt denoted as HT, we utilize this prompt during the HSFT process and the evaluation process. Grounded correction trace prompt is also designed for teacher LLMs to locate and find the error-driven insight from the erroneous reasoning process. And finally, the correction trace prompt is used during our Cross-DPO stage, and further evaluation for self-correction.

## B  DETAILED QUALITATIVE ANALYSIS

In this section, we provide a detailed comparison for error-prone reasoning steps and reasoning results between three different methods, including CoT prompting, our first-stage HSFT models and our SUPERCORRECT.

Table 3: Qualitative comparison between error-prone steps for different methods. Here we use different colors to represents different parts of reasoning. We denote the erroneous reasoning steps in purple, the error cause in red, correct reasoning steps that show improvement in black and the summary for improvement in green.

| Problem | How many subsets of the set of divisors of 72 contain only composite numbers? For example, {8,9} and {4,8,12} are two such sets. **Include the empty set in your count.** (The error-prone parts within the problem) |
|---|---|
| CoT | ## Step 4: Account for the empty set. We need to include the empty set in our count, so we add 1 to the total number of subsets.
(Error Cause: Ignore the fact that empty set has already been counted)
## Step 5: Calculate the final answer. The total number of subsets of the composite numbers is $2^9 + 1 = 512 + 1 = 513$. ✗
(Error Cause: Repeatedly counted empty set) |
| HSFT | ⟨Step4⟩
...... which is $2^9 = 512$.
However, this includes the empty set, which is not a subset of the set of composite numbers.
(Improvement:Notice the empty set is included within the 512 subsets)
⟨/Step4⟩
⟨Step5⟩
We need to subtract 1 from 512 to exclude the empty set. Therefore, the total number of subsets of the set of composite numbers is $512 - 1 = 511$.
✗
⟨/Step5⟩
(Error Cause: Hallucination issue makes LLMs forget the fact that it is required to count empty set in the problem.) |
| **SUPERCORRECT** | ⟨Step6⟩
Calculate the number of subsets containing only composite numbers.
⟨Key⟩
The number of subsets of a set with n elements is $2^n$. Here, with 9 composite divisors, the number of subsets is $2^9 = 512$. This includes the empty set, which is required by the problem statement. ✓
⟨Key⟩
⟨/Step6⟩
Improvement: 1. Successfully notice the empty set is included within 512 subset 2. Mitigate the influence of hallucination issue |

## C    COMPARISON BETWEEN STEP-DPO AND CROSS-MODEL DPO

We conduct qualitative analysis between Step-DPO and our Cross-model DPO. We choose Qwen2.5-Math-Instruct as base model, and we apply Step-DPO on the base model to compare the results. It should be noted that Step-DPO utilize CoT style prompt, for fair comparison, we choose the most suitable prompting method for each model. As shown in Table 4, based on the previous unsolved problems, Step-DPO could locate the error reasoning steps and make corrections (e.g. further identify another multiples of 7), but it but struggles to fully correct them. Compare to Step-DPO, our method could not only locate the erroneous steps, but also conduct accurate self-correction thus solving previous unsolvable problems.

Table 4: Qualitative comparison between Step-DPO and Cross-model DPO.

| Problem | How many of the following numbers are factors of 34 or multiples of 7? Numbers: 1, 2, 3, 4, 8, 14, 17, 29, 56, 91 |
|---|---|
| Base Model | ### Step 1: Factors of 34 The prime factorization of 34 is:

$$34 = 2 \times 17$$

The factors of 34 are:
$$1, 2, 17, 34$$

......
### Step 4: Identify multiples of 7 from the list From the provided list, the multiples of 7 are: - 14 (since $7 \times 2 = 14$)
(Error Cause: Ignore the fact that 56,91 is also multiples of 7)
......
### Step 7: Count the unique numbers The unique numbers that are either factors of 34 or multiples of 7 are: - $1, 2, 14, 17$
Thus, the total count is:
$\boxed{4}$ ✗ |
| Step-DPO | ...... ### Step 2: Identify Multiples of 7 The multiples of 7 are the numbers that can be expressed as $7 \times k$, where $k$ is an integer. Among the given numbers, the multiples of 7 are: - $14 = 7 \times 2$ - $91 = 7 \times 13$
Thus, the multiples of 7 from the list are:

$$14, 91$$

(Error Cause: Still unable to fully recognize all the multiples of 7, missing 56 here)
### Step 3: Combine the Results Now, we combine the factors of 34 and the multiples of 7, ensuring not to double-count any numbers. The unique numbers that are either factors of 34 or multiples of 7 are: - Factors of 34: $1, 2, 17$ - Multiples of 7: $14, 91$
Combining these, we have:

$$1, 2, 14, 17, 91$$

### Conclusion Counting the unique numbers, we find that there are:
$\boxed{5}$ ✗ |
| **Cross-model DPO** | ......
$\langle$Step2$\rangle$
Identify the multiples of 7 from the given list.
$\langle$Key$\rangle$
Multiples of 7 are numbers that can be expressed as 7 multiplied by an integer. It's important to check each number's divisibility by 7 to determine if it qualifies.
Numbers in the list: 1, 2, 3, 4, 8, 14, 17, 29, 56, 91
Multiples of 7: 14, 56, 91
(Self-Correction: Correctly identifies all the multiples of 7)
$\langle$/Step2$\rangle$
$\langle$Step4$\rangle$
Count the total number of unique numbers identified.
$\langle$Key$\rangle$
Counting accurately ensures the final answer reflects the correct quantity of numbers that meet the criteria.
Total numbers: 6
$\langle$/Step4$\rangle$
......
$\langle$Answer$\rangle$ 6 $\langle$/Answer$\rangle$ ✓ |

# D    QUALITY EVALUATION FOR TEACHER LLM GENERATED CONTENT

## D.1    EVALUATION OF INSPECTOR LLM

We discuss the effectiveness of inspector LLM which further ensures the quality of the generated content of Teacher LLMs. As shown in Table 5, we compare the correctness of correction traces generated by three different teacher LLMs across three datasets. The application of the Inspector LLM significantly improves the quality of the final correction traces compared to direct generation. Notably, for LLMs with advanced capabilities that already produce high-quality outputs, it still shows clear improvements. These results demonstrate that the Inspector LLM markedly enhances the accuracy of correction traces, especially for datasets where initial performance was lower.

Table 5: Quantitative analysis of inspector LLM regarding the correctness of correction traces on various datasets.

| Model/Dataset | GSM8K | MATH | GaoKao |
|---|---|---|---|
| Teacher LLM (GPT-4o-mini) | 100% | 92.4% | 89.6% |
| Teacher LLM (GPT-4o-mini) + Inspector LLM (o1-preview) | 100% | **98.8%** | **96.2%** |
| Teacher LLM (GPT-4o) | 100% | 94.4% | 91.3% |
| Teacher LLM (GPT-4o) + Inspector LLM (o1-preview) | 100% | **99.2%** | **97.5%** |
| Teacher LLM (o1-mini) | 100% | 98.2% | 94.8% |
| Teacher LLM (o1-mini) + Inspector LLM (o1-preview) | 100% | **99.6%** | **98.7%** |

## D.2    ANALYSIS ON THE QUALITY OF DIRECT GENERATION

Based on the results in Table 5, the experimental results without the Inspector LLM demonstrate that our directly generated correction traces are already of high quality. We attribute this to our design approach, as outlined below:

- **1. Leveraging Frontier Teacher LLMs:** To ensure the quality of content generated by the teacher LLM, we utilize state-of-the-art LLMs, specifically o1-mini, as the teacher LLM. These models are capable of identifying logical flaws and errors, and they generate high-quality analysis and corrections, as evidenced by the quantitative results.
- **2. Grounding Correction Traces with Ground-Truth Context:** To ensure the accuracy of the correction traces generated by the teacher LLM, as demonstrated in Appendix A, the prompts for generating analysis ($a_i$) and correction ($c_i$) are based on the input question along with the ground-truth solution. This approach grounds the correction trace with the ground-truth solution as context, thereby ensuring the accuracy of the generated content.

# E    MORE ABLATION STUDIES

**Further Analysis on Cross-model DPO**    We first sample 500 erroneous solutions from our dataset, and we use o1-mini to conduct correction trace on the dataset as the ground truth to measure the model alignment (Xu et al., 2022; Zhang et al., 2023b; Khope & Elias, 2022; Guo et al., 2022). We conduct our experiments on three different models after HSFT stage, as shown in Table 6. We additionally introduce two metrics to evaluate the effectiveness of our Cross-model DPO: (1) **Locate correctness**: representing whether the model correctly finds the error steps. (2) **Correction accuracy**: representing whether the model accurately corrects the error steps. We utilize o1-preview as a judger to compare each correction trace generated by the models after Cross-model DPO with the ground truth. From the results, our cross-model DPO shows significant improvements across all models, demonstrating its effectiveness.

Table 6: Quantitative analysis on the effectiveness of our Our Cross-model DPO.

| Model/Metric | Locate correctness | Correction accuracy |
|---|---|---|
| Meta-Llama-3.1 + HSFT | 0.31 | 0.08 |
| Meta-Llama-3.1 + HSFT + Cross-model DPO | **0.49** | **0.27** |
| DeepSeek + HSFT | 0.23 | 0.07 |
| DeepSeek + HSFT + Cross-model DPO | **0.42** | **0.23** |
| Qwen2.5-Math + HSFT | 0.43 | 0.12 |
| Qwen2.5-Math + HSFT+ Cross-model DPO | **0.67** | **0.46** |

**Ablation Study with More Base LLMs**  As shown in Table 7. The result shows that our SuperCorrect can generalize to different LLM architectures, and consistently achieves better performance in both HSFT stage and Cross-model DPO stage, further validating our effectiveness.

Table 7: Ablation study with more base LLMs on MATH and GSM8K. Base1: Llama3.1, Base2: DeepSeek-Math. For simplicity, we denoted Base1-HSFT as **B1-HSFT** and Base2-HSFT as **B2-HSFT**.

| Model | Base1 | Base1+SFT | Base1+HSFT | B1-HSFT+Reflexion | B1-HSFT+Cross-DPO |
|---|---|---|---|---|---|
| MATH (%) | 51.9 | 53.7 | **55.4** | 56.7 | **58.2** |
| GSM8K (%) | 84.5 | 86.2 | **87.2** | 86.8 | **89.7** |
| Model | Base2 | Base2+SFT | Base2+HSFT | B2-HSFT+Reflexion | B2-HSFT+Cross-DPO |
| MATH (%) | 46.8 | 49.2 | **50.9** | 51.2 | **54.6** |
| GSM8K (%) | 82.9 | 84.5 | **85.7** | 85.8 | **88.2** |

**Ablation Study on Prompt Style**  To further evaluate the effectiveness of our meticulously designed hierarchical thought template, we additionally conduct quantitative experiments to show the impact of prompt styles and our hierarchical prompt design. Here we use five prompt styles: 1) CoT 2) CoT + Hierarchical Prompt (without generalization step) 3) CoT + Hierarchical Prompt (with generalization step) 4) Our hierarchical prompt (Not in XML) 5) Our hierarchical prompt (XML). We additionally curated four datasets based on the same 100k math problems with the first four prompt styles. We then trained Qwen2.5-Math-Instruct, Llama3.1-8B-Instruct and DeepSeek-Math-7B on these dataset with the same training settings and evaluate the accuracy on Math dataset. As shown in Table 8, the experimental results indicate that hierarchical reasoning significantly improves model accuracy compared to using CoT as a baseline. Additionally, changing the prompt style (e.g., to XML format) has a small impact on the final accuracy, further demonstrating the effectiveness of our hierarchical reasoning design. Although adding generalization steps helps the model better summarize tasks and thereby enhances its performance, our experimental results indicate that the primary contribution to performance improvements in the HSFT stage comes from the hierarchical reasoning style we designed.

Table 8: Ablation study with different prompt styles. **H** denotes with hierarchical reasoning style and **Gen** denotes with generalization step. **N** denotes not with and **W** denotes with.

| Models | CoT | CoT+H(NGen) | CoT+H(WGen) | Ours(NXML) | Ours(WXML) |
|---|---|---|---|---|---|
| Qwen2.5-Math-7B | 57.4 | 59.7 | 60.8 | 61.8 | 62.4 |
| Llama3.1-8B | 52.6 | 53.3 | 53.6 | 53.7 | 54.1 |
| DeepSeek-Math-7B | 46.8 | 49.6 | 50.2 | 50.6 | 51.6 |

## F  RESULTS OF HIERARCHICAL THOUGHT-BASED REASONING

In this section, we show more detailed hierarchical reasoning process produced by SUPERCORRECT-Qwen-7b on three datasets, including GaoKao, MATH, GSM8K. For each dataset, we present two samples for demonstration. To better present the hierarchical thought during the reasoning process, we denote detailed thought within each step in black, the high-level generalized thought in purple.

**I GaoKao problem:**
Given the function

$$f(x) = \begin{cases} x + 1, & x \leq 0 \\ 2^x, & x > 0 \end{cases}$$

what is the range of values of $x$ that satisfies $f(x) + f\left(x - \frac{1}{2}\right) > 1$?

- - - - - - - - - - - - - - - - - - - - - - - - - - - - - - - - - - - - - - - - -

**Solution Description:**
$\langle$Step 1:$\rangle$
Determine the value of $f(x)$ based on the value of $x$.
$\langle$key$\rangle$
If $x \leq 0$, then $f(x) = x + 1$. If $x > 0$, then $f(x) = 2^x$.
$\langle$/key$\rangle$
$\langle$/Step 1:$\rangle$
$\langle$Step 2:$\rangle$
Determine the value of $f\left(x - \frac{1}{2}\right)$ based on the value of $x$.
$\langle$key$\rangle$
If $x - \frac{1}{2} \leq 0$ (i.e., $x \leq \frac{1}{2}$), then $f\left(x - \frac{1}{2}\right) = \left(x - \frac{1}{2}\right) + 1 = x + \frac{1}{2}$. If $x - \frac{1}{2} > 0$ (i.e., $x > \frac{1}{2}$), then $f\left(x - \frac{1}{2}\right) = 2^{x - \frac{1}{2}}$.
$\langle$/key$\rangle$
$\langle$/Step 2:$\rangle$
$\langle$Step 3:$\rangle$
Analyze the inequality $f(x) + f\left(x - \frac{1}{2}\right) > 1$ in different intervals of $x$.
$\langle$key$\rangle$
This crucial step involves breaking down different cases based on the value of $x$ to apply the correct piecewise definitions of the function.
$\langle$/key$\rangle$
$\langle$/Step 3:$\rangle$
$\langle$Step 4:$\rangle$
Combine the solutions from all cases.
$\langle$key$\rangle$
We find ranges from each case to generate a comprehensive final solution, ensuring we encompass all possible intervals of $x$.
$\langle$/key$\rangle$
$\langle$/Step 4:$\rangle$
$\langle$Generalized$\rangle$
To solve inequalities involving piecewise functions, follow these common steps:
1. **Identify the different cases** based on the definitions of the piecewise function.
2. **Evaluate the function expressions** for each case, including any shifted or transformed inputs.
3. **Set up and solve the inequality** within each interval determined by the cases.
4. **Combine all valid solutions** from each case to form the complete solution set.
This systematic approach ensures accurate and comprehensive solutions to similar problems involving piecewise-defined functions and inequalities.
$\langle$/Generalized$\rangle$
$\langle$Answer:$\rangle$
$x > -\frac{1}{4}$
$\langle$/Answer:$\rangle$

**II GaoKao Problem:**
The secondary product rate of a certain product is 0.02. If you randomly select one from this product, it has been placed back into the pool 100 times. Let $X$ represent the number of secondary products collected, then $D(X) = ?$

---

**Solution Description:**
$\langle$Step 1:$\rangle$
Identify the probability of selecting a secondary product.
$\langle$key$\rangle$
Understanding the probability $p$ is crucial as it forms the basis for calculating the variance. In this case, the secondary product rate is given as 0.02, which means there's a 2% chance of selecting a secondary product each time.
$\langle$/key$\rangle$
$\langle$/Step 1:$\rangle$

$\langle$Step 2:$\rangle$
Determine the number of trials.
$\langle$key$\rangle$
Recognizing the number of trials $n$ is essential. Here, the product has been placed back into the pool 100 times, indicating that each selection is independent and the total number of trials is 100.
$\langle$/key$\rangle$
$\langle$/Step 2:$\rangle$

$\langle$Step 3:$\rangle$
Calculate the expected value $E(X)$ of the number of secondary products collected.
$\langle$key$\rangle$
The expected value $E(X)$ for a binomial distribution is calculated using the formula $E(X) = n \times p$. Substituting the given values:

$$E(X) = 100 \times 0.02 = 2$$

$\langle$/key$\rangle$
$\langle$/Step 3:$\rangle$

$\langle$Step 4:$\rangle$
Calculate the variance $D(X)$ of the number of secondary products collected.
$\langle$key$\rangle$
The variance $D(X)$ for a binomial distribution is given by $D(X) = n \times p \times (1 - p)$. Plugging in the values:

$$D(X) = 100 \times 0.02 \times (1 - 0.02) = 100 \times 0.02 \times 0.98 = 1.96$$

$\langle$/key$\rangle$
$\langle$/Step 4:$\rangle$

$\langle$Generalized:$\rangle$
To solve similar problems involving the number of successes in a fixed number of independent trials with a constant probability of success:
1. **Identify** the probability of success $p$ and the number of trials $n$.
2. **Calculate** the expected value $E(X) = n \times p$.
3. **Determine** the variance $D(X) = n \times p \times (1 - p)$.
This approach uses the properties of the binomial distribution, which is applicable when each trial has two possible outcomes (success or failure) and the probability of success remains constant.
$\langle$/Generalized:$\rangle$

$\langle$Answer:$\rangle$
The variance $D(X)$ of the number of secondary products collected is 1.96.
$\langle$/Answer:$\rangle$

**III MATH Problem:**
What is the tens digit in the sum

$$7! + 8! + 9! + \cdots + 2006!?$$

- - - - - - - - - - - - - - - - - - - - - - - - - - - - - - - - - - - - - - - - - - - -

**Solution Description:**
⟨Step 1:⟩
Identify the problem: We need to find the tens digit in the sum $7! + 8! + 9! + \cdots + 2006!$.
⟨key⟩
Understanding the problem is crucial. We are looking for a specific digit in a large sum, which means we need to focus on the contributions of each factorial to the tens place.
⟨/key⟩
⟨/Step 1:⟩
⟨Step 2:⟩
Recognize that for $n \geq 10$, $n!$ is divisible by 100.
⟨key⟩
This is a key insight because it means that $10!, 11!, 12!, \ldots, 2006!$ do not affect the tens digit of the sum. Only the factorials from 7! to 9! will influence the tens digit.
⟨/key⟩
⟨/Step 2:⟩
⟨Step 3:⟩
Calculate 7!, 8!, and 9!:

$$7! = 5040, \quad 8! = 40320, \quad 9! = 362880$$

⟨key⟩
Calculating factorials can be tedious, but it's essential to ensure accuracy. Each factorial contributes to the overall sum, and we need their last two digits to determine the tens digit of the total sum.
⟨/key⟩
⟨/Step 3:⟩
⟨Step 4:⟩
Extract the last two digits of each factorial:

$$7! \equiv 40 \pmod{100}, \quad 8! \equiv 20 \pmod{100}, \quad 9! \equiv 80 \pmod{100}$$

⟨key⟩
Finding the last two digits involves taking the factorial modulo 100. This simplifies the problem by focusing only on the relevant digits that affect the tens place.
⟨/key⟩
⟨/Step 4:⟩
⟨Step 5:⟩
Sum the last two digits:
$$40 + 20 + 80 = 140$$

⟨key⟩
Adding the last two digits together gives us the intermediate sum. It's important to keep track of the total to determine the final tens digit.
⟨/key⟩
⟨/Step 5:⟩
⟨Step 6:⟩
Determine the tens digit of the sum 140:

$$\text{Tens digit} = 4$$

⟨key⟩
The tens digit is the second digit from the right in a number. Here, it's clear that the tens digit of 140 is 4, which is the answer we seek.
⟨/key⟩
⟨/Step 6:⟩
⟨Generalized:⟩
To solve similar problems, follow these steps: 1. Identify the range of factorials and determine which ones affect the desired digit. 2. Calculate the relevant factorials and extract the necessary digits. 3. Sum these digits and focus on the specific place value required (e.g., tens, hundreds). 4. Extract the desired digit from the final sum.
⟨Generalized:⟩
⟨Answer:⟩
The tens digit is $\boxed{4}$

**IV MATH Problem:**
In how many ways can the letters of the word "COPYRIGHT" be arranged?

------------------------------------------------------------------------

**Solution Description:**
⟨Step 1:⟩
Identify the total number of letters in the word "COPYRIGHT".
⟨key⟩
The word "COPYRIGHT" consists of 9 letters. It's important to count all letters accurately to determine the total arrangements.
⟨/key⟩
⟨/Step 1:⟩
⟨Step 2:⟩
Determine if any letters are repeated in the word.
⟨key⟩
In "COPYRIGHT", all letters are unique. Understanding whether letters are repeated is crucial because repeated letters reduce the number of unique arrangements.
⟨/key⟩
⟨/Step 2:⟩
⟨Step 3:⟩
Apply the formula for permutations of unique objects.
⟨key⟩
Since all letters are unique, the number of arrangements is calculated using the factorial of the total number of letters: 9!.
⟨/key⟩
⟨/Step 3:⟩
⟨Step 4:⟩
Calculate 9! to find the total number of arrangements.
⟨key⟩
$9! = 9 \times 8 \times 7 \times 6 \times 5 \times 4 \times 3 \times 2 \times 1 = 362,880$. This step involves multiplying a series of numbers, which can be calculated step-by-step to avoid errors.
⟨/key⟩
⟨/Step 4:⟩
⟨Generalized:⟩
To determine the number of arrangements of letters in a word, first count the total number of letters. If all letters are unique, use the factorial of the total count. If there are repeated letters, adjust the formula by dividing by the factorial of the count of each repeated letter.
⟨/Generalized:⟩
⟨Answer:⟩
The letters of the word "COPYRIGHT" can be arranged in 362,880 different ways.
⟨/Answer:⟩

**V GSM8K Problem:**

Carlos is planting a lemon tree. The tree will cost $90 to plant. Each year it will grow 7 lemons, which he can sell for $1.5 each. It costs $3 a year to water and feed the tree. How many years will it take before he starts earning money on the lemon tree?

------------------------------------------------------------

**Solution Description:**

⟨Step 1:⟩

Calculate the revenue earned from selling lemons each year.

⟨key⟩

7 lemons * $1.5 per lemon = $10.5.

This step involves multiplying the number of lemons produced by the price per lemon. Understand the importance of identifying fixed production numbers (in this case, 7 lemons) and their respective market price ($1.5). The multiplication here gives us the total income from the lemons before accounting for expenses.

⟨/key⟩

⟨/Step 1:⟩

⟨Step 2:⟩

Calculate the net earnings after deducting annual costs for watering and feeding the tree.

⟨key⟩

$10.5 - $3 = $7.5.

This step is about understanding how to subtract fixed expenses from total revenue to find net income. It's crucial to separate income from costs to ascertain true profit. The $3 cost for watering and feeding is constant each year, impacting the net returns from the lemon sales.

⟨/key⟩

⟨/Step 2:⟩

⟨Step 3:⟩

Determine how many years it takes to cover the initial cost of planting the tree.

⟨key⟩

$90 / $7.5 = 12.

In this step, you're figuring out how long it takes to break even on the initial investment of $90. This involves dividing the total investment by the annual net earnings. Remember that this result indicates the breakeven year, but does not count the year in which the profits actually start.

⟨/key⟩

⟨/Step 3:⟩

⟨Step 4:⟩

Identify the year when he starts earning profit from the lemon tree.

⟨key⟩

12 (years to break even) + 1 = 13.

This final calculation shifts the perspective from a breakeven analysis to profitability. Since he reaches the breakeven point at the end of year 12, he only begins to profit in year 13. This step emphasizes the importance of understanding financial timelines in cash flow analysis.

⟨/key⟩

⟨/Step 4:⟩

⟨Generalized:⟩

The solution involves calculating total income from sales, subtracting operating costs to find net earnings, and determining the break-even point by dividing the initial investment by annual net earnings. Finally, knowing when profit occurs adds critical insight into business investment analysis. For similar problems, follow these steps: identify revenues, calculate net profits, find break-even time, and ascertain the timeline for profitability.

⟨/Generalized⟩

⟨Answer:⟩

13

⟨/Answer⟩

---

**VI GSM8K Problem:**
Tommy is fundraising for his charity by selling brownies for \$3 a slice and cheesecakes for \$4 a slice. If Tommy sells 43 brownies and 23 slices of cheesecake, how much money does Tommy raise?

- - - - - - - - - - - - - - - - - - - - - - - - - - - - - - - - - - - - - - - - - - - - - - -

**Solution Description:**
⟨Step 1:⟩
Calculate the total money raised from selling brownies.
⟨key⟩
To find the total money raised from brownies, we multiply the number of brownies sold by the price per brownie. This is a straightforward multiplication problem.
Here, Tommy sold 43 brownies at \$3 each.
The calculation is: $43 \times 3 = 129$.
Understanding multiplication is crucial as it forms the basis for calculating total revenue from sales.
⟨/key⟩
⟨/Step 1:⟩
⟨Step 2:⟩
Calculate the total money raised from selling cheesecakes.
⟨key⟩
Similar to the previous step, we need to multiply the number of cheesecakes sold by the price per cheesecake.
Tommy sold 23 slices of cheesecake at \$4 each.
The calculation is: $23 \times 4 = 92$.
This step reinforces the concept of multiplication and helps in understanding how to calculate total sales from different products.
⟨/key⟩
⟨/Step 2:⟩
⟨Step 3:⟩
Add the total money raised from both brownies and cheesecakes.
⟨key⟩
Now, we need to combine the total amounts raised from both products to find the overall total.
This involves simple addition: 129 (from brownies) + 92 (from cheesecakes) = 221.
This step is important as it teaches how to aggregate totals from different sources, a common task in finance and fundraising.
⟨/key⟩
⟨/Step 3:⟩
⟨Generalized⟩
To calculate the total funds raised from selling different items, follow these common steps:
1. **Identify the number of items sold** for each product.
2. **Determine the price per item** for each product.
3. **Calculate the total revenue** for each product by multiplying the number of items sold by the price per item.
4. **Sum all individual revenues** to find the overall total funds raised.
This systematic approach ensures accurate calculation of total revenue from multiple sources, which is essential in various financial and fundraising scenarios.
⟨/Generalized⟩
⟨Answer:⟩
Tommy raises \$221.
⟨/Answer⟩

## G IMPROVED SELF-CORRECTION RESULTS

In this section, we select three different self-correction results each from different datasets including MATH, GaoKao, and GSM8K. It should be noted that we split the incorrect reasoning steps with error cause analysis and teacher correction into two parts for better presentation. We denote the error cause in

brown and we denoted the origianl error answer in red, and the correction along with correct answer are denoted in green.

---

**I GSM8K Problem:**

Carlos is planting a lemon tree. The tree will cost $90 to plant. Each year it will grow 7 lemons, which he can sell for $1.5 each. It costs $3 a year to water and feed the tree. How many years will it take before he starts earning money on the lemon tree?

- - - - - - - - - - - - - - - - - - - - - - - - - - - - - - - - - - - - - - - - - - - - - - - - -

**Correct reasoning process before error steps:**

⟨Step 1:⟩

Calculate the total revenue from selling lemons each year.

⟨key⟩

To find the total revenue from selling lemons, multiply the number of lemons by the price per lemon. Here, Carlos sells 7 lemons at $1.5 each. This step is straightforward but crucial as it sets the foundation for understanding the income generated from the lemon tree.

⟨/key⟩

Total Revenue = 7 lemons * $1.5 per lemon = $10.5

⟨/Step 1:⟩

⟨Step 2:⟩

Calculate the total cost of feeding and watering the tree each year.

⟨key⟩

This step involves identifying the annual costs of maintaining the lemon tree. The cost of feeding and watering is $3 per year. Understanding this cost helps in determining the net profit after accounting for all expenses.

⟨/key⟩

Total Cost = $3

⟨/Step 2:⟩

⟨Step 3:⟩

Calculate the net profit per year.

⟨key⟩

Net profit is calculated by subtracting the total costs from the total revenue. This step shows how much money Carlos earns each year after all expenses are accounted for.

⟨/key⟩

Net Profit = Total Revenue - Total Cost = $10.5 - $3 = $7.5

⟨/Step 3:⟩

⟨Step 4:⟩

Identify the initial cost of planting the tree.

⟨key⟩

The initial cost of planting the tree is a one-time expense of $90. This represents the initial investment that needs to be recovered through profits over multiple years.

⟨/key⟩

Initial Cost = $90

⟨/Step 4:⟩

**I Self-Correction For GSM8K Problem:**
**Theses steps below are incorrect in the original reasoning steps**

⟨Step 5:⟩
Calculate the number of years it will take to break even.
⟨key⟩
This step involves dividing the initial investment by the annual net profit to find out how many years it will take for the net profit to cover the initial cost.
⟨/key⟩
Years to Break Even = Initial Cost / Net Profit = \$90 / \$7.5 = 12 years
⟨cause⟩
While this calculation correctly determines that Carlos will break even after 12 years, it does not answer the question of when he starts earning money. The break-even point occurs at the end of the 12th year, meaning he hasn't earned a profit yet within those 12 years.
⟨/cause⟩
⟨/Step 5:⟩ ⟨Step 6:⟩
Determine when Carlos starts earning money from the lemon tree.
⟨key⟩
Since Carlos breaks even at the end of year 12, he will start earning profit from year 13 onwards.
⟨/key⟩
Year when profit starts = Years to Break Even + 1 = 12 + 1 = 13 years
⟨correction⟩
Therefore, Carlos will start earning money on the lemon tree in 13 years, not 12 years.
⟨/correction⟩
⟨/Step 6:⟩
⟨Answer:⟩
13
⟨/Answer:⟩

**II MATH Problem:**
How many different bracelets have 3 identical orange beads, 3 identical black beads, and 1 teal bead, if rotating or flipping a bracelet does not change it?

- - - - - - - - - - - - - - - - - - - - - - - - - - - - - - - - - - - - - - - - - - - - - - - -

**Correct reasoning process before error steps:**
⟨Step 1:⟩
Identify the total number of beads and their types.
⟨key⟩
Understanding the composition of the bracelet is crucial. Here, we have 3 identical orange beads, 3 identical black beads, and 1 teal bead, making a total of 7 beads.
⟨/key⟩
⟨/Step 1:⟩
⟨Step 2:⟩
Calculate the total number of arrangements without considering rotations or reflections.
⟨key⟩
Since the orange and black beads are identical within their groups, we use the formula for permutations of a multiset:
$$\frac{7!}{3! \times 3! \times 1!}.$$
This accounts for the indistinguishability of the identical beads.
⟨/key⟩
Total Arrangements =
$$\frac{7!}{3! \times 3!} = 140.$$

⟨/Step 2:⟩
⟨Step 3:⟩
Apply Burnside's Lemma to account for rotational and reflectional symmetries.
⟨key⟩
Burnside's Lemma is essential in counting distinct arrangements when symmetries (like rotations and reflections) are considered. It requires evaluating the number of arrangements fixed by each symmetry operation and averaging over the total number of symmetries.
⟨/key⟩
⟨/Step 3:⟩
⟨Step 4:⟩
Determine the number of arrangements fixed by each rotation.
⟨key⟩
- **Identity rotation ($0°$)**: All 140 arrangements are fixed.
- **Nontrivial rotations (rotations by $\frac{360°}{7}$ and its multiples)**: No arrangement is fixed because rotating the bracelet changes the positions of the beads, and with the given bead counts, it's impossible to have a configuration that looks the same after such rotations.
⟨/key⟩
Number of fixed arrangements under rotations:
- Identity rotation: 140
- Other rotations: 0
⟨/Step 4:⟩

**II Self-Correction For MATH Problem:**
**Theses steps below are incorrect in original reasoning steps.**

⟨Step 5:⟩
Determine the number of arrangements fixed by each reflection.
⟨key⟩
For reflections, each reflection fixes one bead (the center of reflection) and swaps the remaining beads in pairs. To have an arrangement fixed under reflection, the colors of the beads in each swapped pair must be the same.
⟨/key⟩
⟨cause⟩
With 3 orange beads and 3 black beads (both odd counts), it's impossible to pair all beads of the same color. Each pair requires 2 beads of the same color, so we need even counts of each color. Therefore, no arrangements are fixed under reflections because the color counts do not allow for all swapped pairs to be identical in color.
⟨/cause⟩
Number of fixed arrangements under reflections: 0
⟨/Step 5:⟩
⟨Step 6:⟩
Apply Burnside's Lemma to find the number of distinct bracelets. The original error answer is 20
⟨key⟩
According to Burnside's Lemma, the number of distinct arrangements is the average number of arrangements fixed by each symmetry operation. The total number of symmetries in a bracelet with 7 beads is 14 (7 rotations and 7 reflections).
⟨/key⟩
Calculation:

$$\text{Number of distinct bracelets} = \frac{\text{Total fixed arrangements}}{\text{Number of symmetries}} = \frac{140}{14} = 10.$$

⟨correction⟩
Therefore, the correct number of different bracelets is **10**, not 20. The error in the original calculation was assuming that there were arrangements fixed under reflections, but due to the odd counts of identical beads, no such fixed arrangements exist.
⟨/correction⟩
⟨/Step 6:⟩
⟨Answer:⟩
There are 10 different bracelets.
⟨/Answer:⟩

**III GaoKao Problem:**
Set point $O$ as the origin, the vertical line $x = a$, and the two branches of the hyperbola $C$ : $\frac{x^2}{a^2} - \frac{y^2}{b^2} = 1$ (where $a > 0$ and $b > 0$) intersect at points $D$ and $E$. If the area of quadrilateral $ODE$ is 8, then the minimum value of the distance between the two branches of $C$ is ( )
A. 4
B. 8
C. 16
D. 32

- - - - - - - - - - - - - - - - - - - - - - - - - - - - - - - - - - - - - - -

**Correct reasoning process before error steps:** $\langle$Step 1:$\rangle$
Set point $O$ as the origin and the vertical line $x = a$.
$\langle$/Step 1:$\rangle$
$\langle$Step 2:$\rangle$

Identify the two branches of the hyperbola $C$ : $\frac{x^2}{a^2} - \frac{y^2}{b^2} = 1$, where $a > 0$ and $b > 0$.
$\langle$/Step 2:$\rangle$

**III Self-Correction For GaoKao Problem:**
**Theses steps below are incorrect in original reasoning steps.**

⟨Step 3:⟩
Determine the points $D$ and $E$ where the vertical line $x = a$ intersects the hyperbola $C$.
⟨cause⟩
Substituting $x = a$ into the hyperbola equation yields $y = 0$, resulting in only one intersection point $(a, 0)$. The assumption that there are two distinct points $D(a, b)$ and $E(a, -b)$ is incorrect.
⟨/cause⟩
⟨correction⟩
Substitute $x = a$ into the hyperbola equation:

$$\frac{a^2}{a^2} - \frac{y^2}{b^2} = 1 \implies 1 - \frac{y^2}{b^2} = 1 \implies \frac{y^2}{b^2} = 0 \implies y = 0$$

Therefore, the vertical line $x = a$ intersects the hyperbola $C$ only at the point $D(a, 0)$. Point $E$ does not exist as there is only one intersection point.
⟨/correction⟩
⟨/Step 3:⟩
⟨Step 4:⟩
Calculate the area of quadrilateral $ODE$.
⟨cause⟩
Since there is only one intersection point $D(a, 0)$, the quadrilateral $ODE$ does not exist. The area calculation based on non-existent points $E(a, b)$ and $A(a, -b)$ is invalid.
⟨/cause⟩
⟨correction⟩
Reevaluate the problem statement. If the vertical line $x = a$ intersects the hyperbola at only one point $D(a, 0)$, the area calculation of quadrilateral $ODE$ is not applicable. It is likely that there is a misunderstanding in the problem setup.
⟨/correction⟩
⟨/Step 4:⟩
⟨Step 5:⟩
Find the minimum value of the distance between the two branches of the hyperbola. The original error answer is A = 4
⟨cause⟩
The reasoning that the distance between the two branches is $2a$ is incorrect. Additionally, the approach to minimize $2a$ under the condition $ab = 4$ does not align with the correct geometric interpretation of the hyperbola.
⟨/cause⟩
⟨correction⟩
The distance between the two branches of the hyperbola $C$ is determined by the minimum vertical distance between corresponding points on the branches. To find this, consider the closest points on each branch:

$$\text{Distance} = 2b.$$

Given that the area condition was misapplied, we refer to the correct relationship from the Reference Solution where the minimum distance is found using optimization techniques. The correct minimum distance of the hyperbola $C$ is 8.
⟨/correction⟩
⟨/Step 5:⟩
⟨Answer:⟩
The minimum value of the distance of the hyperbola $C$ is 8.
⟨/Answer:⟩ ✓

