# OpenReview forum: "SuperCorrect: Advancing Small LLM Reasoning with Thought Template Distillation and Self-Correction"
_ICLR.cc/2025/Conference — ICLR 2025 Poster_

### Official Review · Reviewer_hUwP · 2024-10-29

**Soundness:** 3
**Presentation:** 4
**Contribution:** 3
**Rating:** 6
**Confidence:** 4

**Summary:**

The paper introduces SuperCorrect, a two-stage framework to improve the mathematical reasoning of smaller models like Llama-3-8B and DeepSeekMath-Base. While large language models such as GPT-4 and PaLM excel, smaller models struggle with error detection and correction. SuperCorrect addresses this by using a large teacher model to guide the student model through reasoning and reflection. The first stage extracts hierarchical thought templates from the teacher model to refine the student's reasoning. The second stage uses cross-model direct preference optimization (DPO) to enhance the student's self-correction abilities. SuperCorrect-7B achieves state-of-the-art results, outperforming DeepSeekMath-7B and Qwen2.5-Math-7B on MATH and GSM8K benchmarks.

**Strengths:**

1. The writing of the paper is smooth and clear, making it easy to understand.
2. Significant performance improvements were achieved in mathematical tasks.
3. The paper has open-sourced 10K high-quality SFT data and 1K preference alignment data.

**Weaknesses:**

1. The paper introduces two strategies for data augmentation using the teacher model (corresponding to SFT and DPO training), but there is a lack of a clear logical connection between these two methods, as they appear to be separately designed.
2. The motivation behind the design of the "Hierarchical Thought Template" is insufficiently explained. In the Introduction, author mention that the capability deficits in smaller models stem from "failing to effectively identify and correct reasoning errors". However, this doesn't naturally lead to the idea that hierarchical reasoning, incorporating both generalization and details, would be effective.
3. Using a teacher model to correct the student's reasoning results is not a novel idea. Although this paper emphasizes the use of DPO training with these data, there is a possibility that the performance gains primarily come from the expansion of data after correction, and using the SFT approach on this data may achieve similar results.

**Questions:**

1. Are the training data for Qwen2.5-Math-7B-Instruct, DeepSeekMath-7B-Instruct, and Meta-Llama3.1-8B-Instruct the same as the data used for training SUPERCORRECT-7B as mentioned in the paper? Including the types of datasets and the data volumes used.
2. What LLM was used to generate the high-quality dataset ? Considering that LLM can still generate incorrect data (both of Hierarchical Thought process and error correction data), were any measures taken to address this issue?

---

> ### Author Response · Authors · 2024-11-20
> **Response to Reviewer hUwP - Part 1**
>
> *We thank Reviewer hUwP for the constructive review and valuable feedback.We are glad that the reviewer found the writing of our paper to be smooth and clear, and that our proposed framework shows significant performance improvements in mathematical tasks. Please see below for our responses to your comments.*
>
> **Q1: Logical Connection between HSFT and Cross-model DPO.**
>
> A1:
> 1. We did not design our two-stage training paradigm separately. Instead, the HSFT stage and the Cross-model DPO stage are closely interconnected, forming a progressive training paradigm.
> 2. **Guide Student Model to Explicitly Demonstrate Thought Process:** In the Cross-model DPO stage, the teacher LLM is used to correct the  underlying thoughts of students' erroneous self-corrections. Without applying HSFT to the base models, student LLMs struggle to generate coherent thoughts alongside hierarchical reasoning steps, making it hard for cross-model DPO to correct errorneous thoughts. Therefore, the first stage is crucial for enabling student LLMs to reveal their original thoughts behind errors, **serving as correction targets** for the second stage.
> 3. **Align Distribution Between Student and Teacher Models:** As mentioned in our paper, we use teacher LLMs like o1/gpt-4o to generate datasets for both the HSFT and Cross-model DPO stages. However, the training data for Cross-model DPO is initially out-of-distribution (OOD), which can lead to instability, gradient decay, and convergence issues during training. To address this, we aim to align the training model's distribution with that of the teacher LLM during the HSFT stage. By doing so, we can approximate the distribution of HSFT models to match that of the teacher LLM during the Cross-model DPO stage, allowing the teacher-generated data to be treated as in-distribution data for the HSFT models.  Only after thorough training in the HSFT stage can we ensure the quality and stability of Cross-model DPO training.
>
>
> **Q2: Motivation of Hierarchical Reasoning:**
>
> A2:
> 1. Thank you for your kind suggestions. The content that the reviewer mentioned is the motivation for the design of our two-stage framework, SuperCorrect, instead of the motivation of designing hierarchical reasoning.
> 2. As mentioned in L186-196, the motivation of our hierarchical reasoning is inspired from BoT[1]. We found that high-level thought is in-sufficient for solving complex math problems, so we further design hierarchical reasoning to initially improve the reasoning abiltiy of student LLMs. Furthermore, as mentioned in above answer to Q1, utilizing hierarchical reasoning in HSFT stage is to guide the student model to explicitly demonstrate thought process as the correction target for cross-model DPO, which is the second motivation of our design.
>
>
> **Q3: Novelty and Superiority of Our Cross-model DPO.**
>
> A3: We want to emphasize that our Cross-model DPO innovatively focuses on learning to **correct self-correction errors instead of only correcting reasoning errors**, which is a more high-level objective. This allows the model not only to learn the correct solution but also to gain a deeper understanding of the underlying causes of errors from learning correct correction traces.
> Additionally, our Cross-model DPO utilizes a more powerful teacher LLM, and enables the student LLM to learn content beyond its initial distribution towards better teacher LLMs, enabling it to solve previously unsolvable problems.
>
> Besides, we further conduct extensive experiments to show that cross-model DPO is more effective and efficient on self-correction compared to SFT. We used the positive samples from Cross-model DPO as the expansion of correct data for SFT, and we evaluate the model's accuracy on MATH dataset at different checkpoints. As shown in the table below, Cross-model DPO consistently achieves higher accuracy compared to SFT in different training epochs, indicating that simply augmenting data is insufficient to effectively enhance the model's capabilities in this specific scenario.  **In the second stage, the application of Cross-model DPO is the primary reason for the improvement in model performance, which is of better compatibility for self-correction.** Since our objective is to enhance the model's self-correction ability through cross-model corrections, SFT focuses solely on positive samples as learning targets. In contrast, cross-model DPO not only focuses on positive samples by increasing their output probability during optimization but also reduces the output probability of incorrect corrections. This approach **allows the model to understand the correct correction traces while also recognizing the causes of errors**.
>
> | In Second Stage    | Epoch=2 | Epoch=4 | Epoch=6 | Epoch=8 |
> | --------------- | ------- | ------- | ------- | ------- |
> | SFT        | 62.8    | 63.1    | 63.7    | 64.1    |
> | Cross-model DPO | 62.9    | 64.5    | 67.8    | 70.2    |

---

> ### Author Response · Authors · 2024-11-20
> **Response to Reviewer hUwP - Part 2**
>
> **Q4: Training Setup for SuperCorrect-7B Models.**
>
> A4:
> 1. **Clarification of Models:**  As mentioned in L94-97 and L404-442, we apply SuperCorrect to three different base models: Qwen2.5-Math-7B-Instruct, DeepSeekMath-7B-Instruct, and Meta-Llama3.1-8B-Instruct, collectively referred to as SuperCorrect-Qwen/DeepSeek/Llama-7B. We selected the model with the best performance, SuperCorrect-Qwen-7B, and designated it as SuperCorrect-7B.
>
> 2. **Clarification of Training Data:** Since Qwen2.5-Math-7B-Instruct, DeepSeekMath-7B-Instruct, and Meta-Llama3.1-8B-Instruct are proposed by three different companies, the training data for these three models are obviously different. We assume you are asking whether SuperCorrect-Qwen/DeepSeek/Llama-7B were trained with the same data. In response to this question, as mentioned in L402-444, for all these three models, we use the same training data in HSFT stage and Cross-model DPO stage, as well as the same training setup, including all hyperparameters.
>
>
> **Q5: Provide LLMs that Generate High-quality Dataset and Measures to Ensure the Quality.**
>
> A5: As mentioned in L196-201, we utilize frontier LLMs such as o1-mini to generate a high-quality dataset, which ensures the quality of the generated hierarchical reasoning and correction traces. **We have updated our manuscript in Appendix.D for more detailed analysis.** Here we summarize our measures to ensure the generation quality in three aspects:
>
> *  **Iterative Evaluation to Ensure Quality** : To address potential errors in the generated content, our method incorporates an evaluation process during the dataset curation phase. We utilize an inspector LLM, which verifies the accuracy of the correction trace by comparing it against the input problem and the ground-truth solution. If issues are detected, the problematic parts are sent back to the teacher LLMs for revision. This iterative checking process continues until no errors remain, with a maximum of three iterations allowed. We present both quantitative and qualitative results of our evaluation method. The table below compares the correctness of correction traces generated by three different teacher LLMs across three datasets. The application of the Inspector LLM significantly improves the quality of the final correction traces compared to direct generation. Notably, even for LLMs with advanced capabilities that could already produce high-quality outputs, the Inspector LLM still brings clear improvements. These results demonstrate that the Inspector LLM markedly enhances the accuracy of correction traces, especially for datasets where initial performance was lower. **This iterative evaluation method effectively ensures the quality and reliability of the generated content.**
>
> | Model/Dataset                                           | GSM8K | MATH  | GaoKao |
> | ------------------------------------------------------- | ----- | ----- | ------ |
> | Teacher LLM (GPT-4o-mini)                               | 100%  | 92.4% | 89.6%  |
> | Teacher LLM (GPT-4o-mini)  + Inspector LLM (o1-preview) | 100%  | **98.8%** | **96.2%**  |
> | Teacher LLM (GPT-4o)                                    | 100%  | 94.4% | 91.3%  |
> | Teacher LLM (GPT-4o)  + Inspector LLM (o1-preview)      | 100%  | **99.2%** | **97.5%**  |
> | Teacher LLM (o1-mini)                                   | 100%  | 98.2% | 94.8%  |
> | Teacher LLM (o1-mini)  + Inspector LLM (o1-preview)     | 100%  | **99.6%** | **98.7%**  |
>
> * **Leveraging Frontier Teacher LLMs:** To ensure the quality of teacher LLM generated content, we leverage state-of-the-art LLMs such as o1-mini as teacher LLM, which are capable of finding logic flaws/errors and generating high-quality analysis and correction as shown in the quantitative results.
>
> * **Grounding Correction Traces with Ground-Truth Context:** To ensure the accuracy of the correction trace generated by the teacher LLM, as demonstrated in Appendix A, the prompt for generating analysis ($a_i$) and correction ($c_i$) is based on the input question along with the ground-truth solution. This approach grounds the correction trace with the ground-truth solution as context, thereby guaranteeing the accuracy of the generated content.
>
> [1] Yang L, Yu Z, Zhang T, et al. Buffer of Thoughts: Thought-Augmented Reasoning with Large Language Models. NeurIPS 2024.

---

> ### Author Response · Authors · 2024-11-24
> **Gentle Reminder**
>
> Dear reviewer hUwP:
>
> We sincerely appreciate the time and effort you dedicated to reviewing our paper. In response to your concerns, we have conducted additional experiments and provided an in-depth analysis on our method during the discussion period. We summarize our responses to each of your questions as follows:
>
> 1. We clarified the logical connection between HSFT and Cross-model DPO and emphasized the motivation behind our hierarchical reasoning.
> 2. We highlighted and explained the novelty of our SuperCorrect, which focuses on learning to correct self-correction errors rather than just reasoning errors. We further conducted additional quantitative experiments to demonstrate the superiority of our Cross-model DPO.
> 3. We provided detailed clarification of the training models and data.
> 4. We provided more information about the teacher LLMs and conducted quantitative experiments. We also provided a detailed analysis of our evaluation measurement and generation design, which ensures the quality and reliability of the generated content.
>
> As the discussion period concludes in two days, we kindly request, if possible,  that you review our rebuttal at your convenience. Should there be any further points requiring clarification or improvement, please know that we are fully committed to addressing them promptly. Thank you once again for your invaluable contribution to our research.
>
> Warm regards,
>
> The Authors

---

> > ### Comment · Reviewer_hUwP · 2024-11-26
> >
> > Thanks for the reponse!
> >
> > 1. I understand the importance of HSFT for mathematical tasks, which also explains why this method achieves more significant performance improvements in math.
> > However, my main concern is why generating a hierarchical reasoning process is critical for the subsequent "correct self-correction errors." If the goal is merely to "demonstrate the thought process," a standard CoT approach could achieve the same purpose.
> >
> > 2.I think "correct self-correction errors instead of only correcting reasoning errors" is a good insight. Could you further explain why cross-model DPO can achieve this goal? Providing specific examples, such as cases from the training data, would make it even clearer.

---

> ### Author Response · Authors · 2024-11-26
> **Response to reviewer hUwP for addressing remaining concerns**
>
> We sincerely thank you for your prompt feedback and discussion. In response to your remaining concerns, we further make illustrative clarification of these two questions:
> * For question1: we have to clarify that compared to traditional CoT, our HSFT is to not only demonstrate the thought process, but also **provide more self-correction targets for more informative and fine-grained self-correction feedbacks in cross-model DPO process**.
>   * Qualitatively speaking, standard CoT approach only generates detailed explanation for each step and fails to produce high-level thought which serves as a generalized template to solve similar problems. Our hierarchical reasoning incorporates both high-level thought and detailed thought, which can enable cross-model DPO to **identify more comprehensive and fine-grained errors in reasoning process at hierarchical levels**, allowing for better self-correction.
>   * We also conducted experiments (Table.8 in Appendix.E) to showcase the superiority of our hierarchical SFT. We list some results in the table below:
> | Models/Prompt Style | CoT  | CoT + Hierarchical |
> | ------------------- | ---- | ---------- |
> | Qwen2.5-Math-7B     | 57.4 | **60.8**       |
> | Llama3.1-8B         | 52.6 | **53.6**      |
> | DeepSeek-Math-7B    | 46.8 | **50.2**       |
> The experimental results indicate that our hierarchical reasoning significantly improves model's final reasoning accuracy compared to CoT, further demonstrating the importance of our HSFT in identifying self-correction errors.
>
> * For question2: Traditional methods only focus on correcting reasoning errors (e.g., Step-DPO), which construct **paired correct and wrong reasoning processes** in DPO stage for optimizing LLM's reasoning ability.  In contrast, our SuperCorrect focuses on correcting self-correction errors, which innovatively constructs **paired correct and wrong correction traces** in cross-model DPO stage for optimizing student LLMs' self-correction ability towards teacher LLMs' correction ability.
> Here, we provide detailed illustrative comparisons between previous paired reasoning processes and our paired correction traces for better understanding:
>
> ### ```Paired Correct and Wrong Reasoning Processes```
>
> **Input:**
> Let $f(x)=\frac{1}{2x-5}$. Find the largest $x$ which is not in the domain of $g(x)=f(f(x))$.
>
> ### **Correct reasoning process**
>
> #### Step 1:
> ......
>
> #### Step 4: (Correct Simplification)
> Simplify the expression for $g(x)$.
>
> $$
> g(x)=\frac{1}{\frac{1}{x-2.5}-5}=\frac{x-2.5}{1-5(x-2.5)}=\frac{x-2.5}{13.5-5x}
> $$
>
> ### **Wrong reasoning process**
>
>
> #### Step 1:
> ......
>
> #### Step 4: (Error Reasoning Step)
> Simplify the expression for $g(x)$.
>
> $$
> g(x)=\frac{1}{\frac{1}{x-2.5}-5}=\frac{1}{\frac{1-5x}{x-2.5}}=\frac{x-2.5}{1-5x}
> $$
>
>
> ### ```Paired Correct and Wrong Correction Traces (Ours)```
>
> **Input:**
> Let $f(x)=\frac{1}{2x-5}$. Find the largest $x$ which is not in the domain of $g(x)=f(f(x))$.
> #### Step 1:
> ......
>
> #### Step 4: (Error Reasoning Step)
> Simplify the expression for $g(x)$.
>
> $$
> g(x)=\frac{1}{\frac{1}{x-2.5}-5}=\frac{1}{\frac{1-5x}{x-2.5}}=\frac{x-2.5}{1-5x}
> $$
>
> ### **Correct correction trace**
>
> #### Step 4 (Error located)
>
> Simplify the expression for $g(x)$.
>
> $$
> g(x) = \frac{1}{\frac{1}{x-2.5} - 5} = \frac{1}{\frac{1 - 5(x-2.5)}{x-2.5}} = \frac{x-2.5}{1 - 5(x-2.5)}
> $$
>
> **Cause**
>
> The simplification in the original reasoning incorrectly reduced $\frac{1}{x-2.5} - 5$ to $\frac{1 - 5x}{x-2.5}$. The correct simplification should account for distributing the negative sign properly.
>
> **Correction**
>
> $$
> g(x) = \frac{1}{\frac{1}{x-2.5} - 5} = \frac{1}{\frac{1 - 5(x-2.5)}{x-2.5}} = \frac{x-2.5}{1 - 5(x-2.5)} = \frac{x-2.5}{1 - 5x + 12.5} = \frac{x-2.5}{13.5 - 5x}
> $$
>
> ### **Wrong correction trace**
>
> #### Step 4: (Error overlooked)
> Simplify the expression for $g(x)$.
>
> $$
> g(x)=\frac{1}{\frac{1}{x-2.5}-5}=\frac{1}{\frac{1-5x}{x-2.5}}=\frac{x-2.5}{1-5x}
> $$
>
> **Evaluate**
>
> This step is correct.
>
>
> We hope our response helps you better understand our method and addresses your concerns. If you have any further questions, please feel free to reach out for further discussion.

---

> > ### Comment · Reviewer_hUwP · 2024-11-27
> >
> > I believe the advantages of HSFT  can be further explored. Beyond its ability to demonstrate fine-grained reasoning, the more intriguing aspect lies in its distinct thought processes and reasoning workflows compared to CoT. Additionally, I suggest that the authors include some case studies in the appendix to visually illustrate the high-level capability of "correct self-correction errors."Overall, the authors' response addressed most of my concerns, and I will increase my score.

---

> > > ### Author Response · Authors · 2024-11-27
> > > **Thanks for your support**
> > >
> > > Dear Reviewer hUwP，
> > >
> > > Thank you for raising score! We greatly appreciate your recognition of our work and the valuable feedback you provided. We will continue to optimize this method and include more better case studies in final version.
> > >
> > > Warm Regards,
> > >
> > > The Authors

---

### Official Review · Reviewer_gRHY · 2024-10-31

**Soundness:** 3
**Presentation:** 3
**Contribution:** 3
**Rating:** 6
**Confidence:** 3

**Summary:**

The paper introduces SUPERCORRECT, a two-stage framework aimed at boosting smaller LLMs' reasoning and self-correction abilities. It leverages guidance from larger teacher models, such as GPT-4 or LLaMA, in two core phases: hierarchical thought templates and cross-model collaborative Direct Preference Optimization. Experiments on MATH and GSM8K benchmarks reveal that SUPERCORRECT consistently outperforms traditional reflection and fine-tuning approaches. Notably, the SUPERCORRECT-7B model exceeded DeepSeekMath-7B by 7.8% on MATH and 5.3% on GSM8K, showcasing improvements in both accuracy and stability.

**Strengths:**

1. By utilizing the teacher model to identify and correct errors in the student model's reasoning, SUPERCORRECT not only corrects mistakes but also teaches the student model to avoid and rectify specific errors. This approach breaks the bottleneck of the student model's thought process and equips it with new skills and knowledge to tackle challenging problems.

2. When leverage a large teacher model to supervise and correct a smaller student model, SUPERCORRECT combines hierarchical thought templates and cross-model collaborative direct preference optimization (DPO). I think this method is innovative.

3.  This advancement is particularly significant as it shows the potential for smaller models to compete with or even surpass the capabilities of much larger models in complex mathematical reasoning tasks.

**Weaknesses:**

1. The paper primarily focuses on 7B models. It may not be immediately clear how well the SUPERCORRECT framework would scale to larger models or generalize across different types of reasoning tasks beyond mathematical problems.

2. The success of SUPERCORRECT relies heavily on the quality of the fine-tuning datasets and the paired correction traces. The paper mentions constructing high-quality datasets, but it may face challenges in scenarios where such curated datasets are not available or the domain of interest is very niche.

**Questions:**

1.	Could you elaborate on the potential strategies for scaling the SUPERCORRECT framework to even larger language models, and how you might address the computational efficiency challenges that come with such an increase in model size?

2.	How does SUPERCORRECT handle systematic biases or errors that may be present in the teacher model's corrections? Additionally, could you discuss how the framework evaluates and ensures the robustness of the student model against such potential inaccuracies in the supervision process?

3.	In the context of the two-stage training process, have you observed any stability issues or challenges in long-term training? If so, what techniques or modifications are employed to ensure the stability and convergence of the models?

---

> ### Author Response · Authors · 2024-11-20
> **Response to Reviewer gRHY - Part 1**
>
> *We thank Reviewer gRHY for the positive review and valuable feedback. We are glad that the reviewer found the proposed framework to be innovative and that it outperforms traditional approaches. Additionally, we are pleased that the reviewer noted its ability to greatly improve the student model's performance. Please see below for our responses to your comments.*
>
> **Q1: Scale SuperCorrect to Larger Sized Models.**
>
> A1:
> 1. Thank you for your insightful suggestions. The primary goal of our work is to introduce a novel training paradigm that enhances small-sized LLMs' abilities to conduct self-correction on erroneous reasoning steps and solve mathematical problems. However, our paradigm can also be applied to a broader range of tasks and larger-sized models.
>
>
> 2. We have the plan to generalize our method to larger-sized LLMs and more kind of tasks. However, due to the demanding requirements of training larger-sized LLMs, which require more computational resources, it cannot be achieved in a short period of time.
>
> 3. We have applied our method to wider range of tasks, and the results are promising. Thank you again for you kind suggestion, and please stay tuned for our updates.
>
> **Q2: Challenges of Curatinng Datsets on Specific Domain.**
>
> A2:
> 1. Thank you for your insightful advice. The challenges that you mentioned are common problems faced by all LLMs and subsequent works. This is not the specific limitation of our method.
> 2. The core concept of our method is to further improve the reasoning and self-correction ability of LLMs based on an well-established domains such as Math, Physics, Medical etc. For LLMs that already possess basic knowledge of a specific domain, our method could theoretically show significant improvements.
> 3. To overcome the challenge you mentioned, we can synthesize data in these specific domains using sota LLMs and utilize advanced OCR technology to scan documents for dataset creation. Additionally, we can extend the open-source datasets with the help of powerful LLMs to address data-related challenges.
>
>
> **Q3：Potential Strategy for Scaling.**
>
> A3:
> Thank you for your kind reminder. We have already considered using different strategies to scale our models to larger sizes and improve computational efficiency. We have already explored common methods for LLMs as follows:
> * **1. Enhanced Training Frameworks:** Utilizing advanced training frameworks such as DeepSpeed and vLLM could help to improve both performance and efficiency. For example, we can utilize vLLM’s PagedAttention to efficiently manage KV caches, reducing memory bottlenecks and increasing throughput by 14-24 times compared to traditional methods. We can also use tensor parallelism to distribute computations across multiple GPUs, maximizing resource utilization and speeding up inference.
>
> In addition, our approach may encounter two main issues when scaling: 1) As the model size increases, generating a larger volume of high-quality data introduces greater generation overhead. 2) With the increase in model size, more parameters require additional computational resources and training time. To solve these specific problems of our methods, we propose two possible ways:
> * **2. Effcient Generation through Speculative Decoding:** Based on the fact that we have to utilize LLMs to generate larger-sized high-quality dataset for training, we can use speculative decoding to improve inference efficiency of teacher LLMs. Specifically, it means that we can sue use a smaller-sized teacher LLM to predict multiple tokens autoregressively. The larger-sized teacher LLM then verifies these tokens in parallel to check if the outputs align with expectations. If they do, the tokens are accepted; if not, corrections are made. This approach accelerates inference by combining the efficiency of the smaller model with the accuracy of the larger one, which could efficiently speed up our data curation process.
> * **3. Selectively Choose More Important Tokens:** With the scaling of model parameters and dataset size, it is costly to training on all the tokens especially on larger-sized models. So we can design a loss-related mechanism to filter those harmful or useless tokens during training process, which could help to improve the training efficiency and may lead to further improvement of the performance.

---

> ### Author Response · Authors · 2024-11-20
> **Response to Reviewer gRHY - Part 2**
>
> **Q4: How to handle systematic biases or errors?**
>
> A4:
> 1. To address systematic biases or errors in the teacher model's corrections, our method employs a strict validation process during dataset curation. We use an inspector LLM, which cross-verifies the teacher's generated corrections against the input problem and correct solution. This process identifies any inaccuracies, and problematic parts are sent back to the teacher LLM for revision. This cycle is repeated up to three times to ensure thorough error-checking. If errors persist, human annotators step in to provide the correction trace.
>
> 2. What's more, our approach ensures accuracy by grounding the teacher's generated content with ground-truth solution. We utilize state-of-the-art LLMs, like o1-mini, known for their ability to detect logical flaws and produce high-quality analysis. This approach minimizes the impact of potential inaccuracies in the supervision process, thereby enhancing the student model's reliability and performance.
>
> 3. While the detailed process is not included in the original version due to space constraints, our experimental results indicate the high quality and reliability of the generated content. **We have updated our manuscript, and present the relevant experiments and analysis in Appendix.D.**
>
>
> **Q5: Techniques and Modifications to Ensure the Stability and Convergence during Training.**
>
> A5: We observed some stability issues and challenges during long-term training. We find that the stability of training in the Cross-model DPO stage largely depends on the quality of training in the HSFT stage. **Importance of Distribution Alignment:** Through our numerous experiments, we observed that inadequate training during the HSFT stage, or not using the same teacher LLM as in the Cross-model DPO stage, can lead to instability, gradient decay and convergence issues during the Cross-model training process. We attribute this to the out-of-distribution (OOD) issue.
>
> To address this, we need to align the distribution of the training model with the teacher LLM in the HSFT stage as much as possible. This alignment allows us to approximate the distribution of HSFT models to that of the teacher LLM, treating the teacher LLM-generated data as in-distribution, self-generated data for the HSFT models. Based on this analysis, we further curated a larger dataset containing 100K HSFT data, and increase the training epochs to ensure the alignment between HSFT models with teacher LLM. After implementing these optimizations, the training process in the Cross-model DPO stage became more stable and achieved faster convergence.

---

> > ### Comment · Reviewer_gRHY · 2024-11-27
> > **Response**
> >
> > Thanks for the authors' responses to address my questions.
> > I have the following question: I checked Table 1's results in the paper. I found the reported result of "Qwen2.5-Math-7B-Instruct" is not consistent with the model from the original paper by "Yang et al., 2024a: "
> >
> > An Yang, Beichen Zhang, Binyuan Hui, Bofei Gao, Bowen Yu, Chengpeng Li, Dayiheng Liu, Jianhong Tu,
> > Jingren Zhou, Junyang Lin, et al. Qwen2. 5-math technical report: Toward mathematical expert model via
> > self-improvement. arXiv preprint arXiv:2409.12122, 2024a.
> >
> > Can you explain why causing this inconsistency? Is the experimental comparison fair? I cannot check every result in the paper, but we should pay more attention to the results.

---

> ### Author Response · Authors · 2024-11-27
> **Response to Reviewer gRHY**
>
> We sincerely thank you for your feedback. In response to your concern about the evaluation results, we will explain the experimental results in our paper as the following:
>
> 1. As indicated in the caption of Table 1, the performance of Qwen2.5-Math-7B-Instruct was assessed using an open-source evaluation framework [3], diverging from the evaluation methodology described in [1]. We want to emphasize that the results for Qwen2.5-Math-7B-Instruct reported in [1] were obtained using a distinct evaluation approach [2]. This approach significantly differs from the widely recognized open-source frameworks (e.g., lm-evaluation [3]) prevalent within the Large Language Model (LLM) community. Moreover, Qwen2.5-Math [1] incorporates additional techniques/tricks during its evaluation process, resulting in notably higher accuracy than that typically achieved through standard evaluation frameworks.
>
> 2. It is noted that the evaluation of existing LLM models like Meta-Llama3.1 [4], DeepSeek [5], and GPT-4 [6] predominantly utilizes these broadly accepted open-source frameworks rather than the method outlined in [2]. Therefore, we **aimed for a fair comparison**, as detailed in Table 1, both Qwen2.5-Math-7B-Instruct and SuperCorrect-Qwen-7B **were evaluated using the same lm-evaluation [3] framework**. The final results sufficiently demonstrate the superiority of our SuperCorrect over all previous methods.
>
> We hope our response helps you better understand our method. If you have any further questions, please feel free to reach out for further discussion.
>
> [1] An Yang, Beichen Zhang, Binyuan Hui, Bofei Gao, Bowen Yu, Chengpeng Li, Dayiheng Liu, Jianhong Tu, Jingren Zhou, Junyang Lin, et al. Qwen2. 5-math technical report: Toward mathematical expert model via self-improvement. arXiv preprint arXiv:2409.12122, 2024a.
>
> [2] https://github.com/QwenLM/Qwen2.5-Math
>
> [3] https://github.com/EleutherAI/lm-evaluation-harness
>
> [4] Dubey A, Jauhri A, Pandey A, et al. The llama 3 herd of models[J]. arXiv preprint arXiv:2407.21783, 2024.
>
> [5] Shao Z, Wang P, Zhu Q, et al. Deepseekmath: Pushing the limits of mathematical reasoning in open language models[J]. arXiv preprint arXiv:2402.03300, 2024.
>
> [6] Achiam J, Adler S, Agarwal S, et al. Gpt-4 technical report[J]. arXiv preprint arXiv:2303.08774, 2023.

---

### Official Review · Reviewer_Bwuv · 2024-11-03

**Soundness:** 3
**Presentation:** 4
**Contribution:** 3
**Rating:** 6
**Confidence:** 2

**Summary:**

The paper tries to improve the mathematical reasoning of small LLMs by (1) improving their thought template using their proposed Hierarchical Thought  template and (2) enhancing the self-correction of the model by making it to learn how to self-correct like a larger SOTA model using the proposed cross-model DPO. The experiments are done over three small LLMs on two math benchmarks.

**Strengths:**

**(a)** I think the paper is very well-written and even someone who is not entirely familiar with self-correction and reflection concepts can follow. The formatting is also clean and makes the story again easier to follow. The preliminary concepts such as DPO and how it relates to other methods such as RLHF is again very well presented.

**(b)** The cross-model DPO, to me, seems quite close to a distillation framework of teaching the smaller model how to identify wrong steps from a larger model. I think it is very simple and quite applicable to many domains with different tasks (see also Question **(a)** below).

**(c)** The quantitative results seem quite promising, especially in Table 1 and 2. Both HTSF and cross-model DPO seem to give significant gains. (see also weakness **(c)** below)

**Weaknesses:**

**(a)**  While I find the quantitative results quite promising, I think there are a few tiny tables that can be added to further prove the point and support the proposed intuition. Specifically, the way I see it, is that the self-correction ability of the smaller (student) LLM is improved by comparing it with the correction provided from a larger, more powerful teacher, as if the paper is ‘distilling’ the self-correction. While it seems to work well for final accuracy, it is not quantitatively supported that this distillation was successful. For example, I would like to see if the two models agree on each reasoning step after the distillation and can locate correct and wrong steps the same way. This should be quantifiable, as you already have discrete reasoning steps in XML format. Specifically, I suppose you can measure the ratio of steps that are predicted the same way (both larger LM and smaller LM) say `correct’ or `wrong’ over the entire number of steps. Then, based on what the paper claims, I expect this ratio to increase after the cross-model collaborative DPO stage. I let the authors decide what sort of metrics demonstrates this the best, but I think right now such an inspection is missing.


**(b)** Regarding the Thought Template, the main contribution seems to be the hierarchical aspect of the thought. However, I think this needs to be supported by measuring variance over different prompt styles, while I would maintain the hierarchical aspect. For example, would we get the same benefits if the <Generalized> statement is moved to be before all of the <Step>s? (I understand if one works better than the other, but overall I would expect this should still give similar gains?) The other way to test this would be to only remove the Generalized prompt (or negatively prompt the model to not provide this). Or you could even use the same exact prompt as the baseline, and only improve it with a statement to include the <Generalized> template as well. In short, I’m currently finding the gains from HSFT alone, compared to SFT, hard to associate **only** to the hierarchical aspect and not to the different prompt formatting used.

**(c)** I would like to ask the authors to clarify why Table 2 only shows numbers for the Qwen model and not for the DeepSeek and Llama, as this would not require any additional experiments!? In the current version the individual gains for HSFT(without cross-model DPO) is only demonstrated for Qwen and not the other two models.

**Questions:**

**(a)** It is stated multiple times that self-correction of reasoning steps is additionally difficult for mathematical tasks. While I mostly agree with this, the current proposed method has no part in its design that limits itself to Math benchmarks; the proposed Hierarchical Thought Template, to some extent, can be used for any task and the cross-model correction with a teacher model can also be used in other tasks. Therefore, it currently seems strange to me that the paper only evaluates on the math benchmarks and only for smaller LM models tuned for math questions. I would like to ask the authors to clarify this.

---

> ### Author Response · Authors · 2024-11-20
> **Response to Reviewer Bwuv - Part 1**
>
> *We thank Reviewer Bwuv for the positive review and valuable feedback. We are glad that the reviewer found that our paper is well-written and well-presented, the proposed framework is generalizable and promising, our SuperCorrect-7B shows significant gains. Please see below for our responses to your comments.*
>
> **Q1: Quantitative Analysis on the Effectiveness of our Our Cross-model DPO.**
>
> **A1:**
> In response to the reviewer's concern, to quantitatively assess the effectiveness of our Cross-model DPO, we focus on the concept of cross-model alignment. To achieve this, we first **sample 500 erroneous solutions from our dataset**, and we use o1-mini to conduct correction trace on the dataset as the ground truth to measure the model alignment. We conduct our experiments on three different models after HSFT stage, as shown in the table. We additionally introduce two factors to evaluate the effectiveness of our Cross-model DPO: (1) Locate correctness, representing whether the model correctly finds the error steps. (2) Correction accuracy, representing whether the model accurately corrects the error steps. We utilize o1-preview as a judger to compare each correction trace generated by the models after Cross-model DPO with the ground truth. As shown in the table below, both factors show significant improvement across all models, demonstrating the effectiveness of our Cross-model DPO and improved cross-model alignment.
>
> | Model/Factor     | Locate correctness | Correction accuracy |
> | --- | ---- | ----- |
> | Meta-Llama-3.1 + HSFT    | 0.31 | 0.08 |
> | Meta-Llama-3.1 + HSFT + Cross-model DPO | **0.49** | **0.27**  |
> | DeepSeek + HSFT   | 0.23  | 0.07   |
> | DeepSeek + HSFT + Cross-model DPO | **0.42**  | **0.23** |
> | Qwen2.5-Math + HSFT  | 0.43 | 0.12 |
> | Qwen2.5-Math + HSFT+ Cross-model DPO | **0.67** | **0.46** |
>
>
>
> **Q2: Ablation study on Prompt Style.**
>
> **A2:**
> In response to reviewer's concern, we conduct additional experiments to show the impact of prompt styles and our hierarchical prompt design. Here we use three prompt styles: 1) CoT 2) Hierarchical Prompt (Not in XML) 3) Hierarchical Prompt (XML). We additionally curated two datasets based on the same 100k math problems with the first two prompt styles. We then trained Qwen2.5-Math-Instruct, Llama3.1-8B-Instruct and DeepSeek-Math-7B on these dataset with the same training settings and evaluate the accuracy on Math dataset. The results are showed in the table below. The experimental results indicate that hierarchical reasoning significantly improves the accuracy of model after training compared to CoT as a baseline. Additionally, changing the prompt style (e.g., to XML format) has a small impact on the final accuracy, thereby demonstrating the effectiveness of our hierarchical reasoning design.
> | Models/Prompt Style | CoT  | Hierarchical Prompt （Not XML） | Hierarchical Prompt (XML) |
> | ------- | :----: | :---:| :----: |
> | Qwen2.5-Math-7B-Instruct | 57.4 | 61.8 | 62.4|
> | Llama3.1-8B-Instruct     | 52.6 | 53.7  | 54.1|
> | DeepSeek-Math-7B  | 46.8 | 50.6 | 51.6|
>
> Here we present the prompt we used in the experiments:
>
> **CoT:**
>
> Please reason step by step, and put your final answer within \\boxed{}.
>
> **Hierarchical prompt (Not in XML format):**
>
> **Task:** Solve the following math problem step-by-step. Each step should be clearly numbered (e.g., Step 1, Step 2).
>
> 1. **Identify the Problem:** Begin by stating the problem clearly.
>
> 2. **Step-by-Step Breakdown:** For each step:
>    - Clearly describe the action you are taking.
>    - Assess whether this step is challenging or tricky. If it is, provide a detailed explanation and analysis to help clarify your reasoning process.
>
> 3. **Combine Results:** After identifying all relevant factors or multiples, compile the results carefully, ensuring no duplicates are counted.
>
> 4. **Count Unique Results:** Count the total number of unique results identified from the previous steps.
>
> 5. **Generalization:** After completing the solution, summarize the common solution methods and reasoning steps. This summary should help you and your classmates who struggle with math to generalize these techniques to similar problems.
>
> 6. **Final Answer:** Present the final answer clearly at the end.
>
>
> **Hierarchical Prompt(XML format):**
>
> Solve the following math problem in a step-by-step XML format, each step should be enclosed within tags like <Step1></Step1>. For each step enclosed within the tags, determine if this step is challenging and tricky, if so, add detailed explanation and analysis enclosed within <Key> </Key> in this step, as helpful annotations to help you think and remind yourself how to conduct reasoning correctly. After all the reasoning steps, summarize the common solution and reasoning steps to help you and your classmates who are not good at math generalize to similar problems within <Generalized></Generalized>. Finally present the final answer within <Answer> </Answer>.

---

> ### Author Response · Authors · 2024-11-20
> **Response to Reviewer Bwuv - Part 2**
>
> **Q3: More Quantitative Results on Table 2.**
>
> **A3:**
> We conducted quantitative experiments on all our models. However, due to page limits, we only selected the model with the best performance as our base model, as shown in Table 2. In response to your concern, we will provide the rest of the experiments in the table below. The result shows that our method consistently achieves better performance in both HSFT stage and Cross-model DPO stage, further validating the effectiveness of our approach.
>
> | **Model**     | Llama3.1 | Llama3.1 + SFT | Llama3.1 + HSFT | Llama3.1-HSFT + Reflexion | Llama3.1-HSFT + Cross-DPO |
> | ------------- | :------:  |:------:  | :------:  |:------:  | :------: |
> | **MATH (%)**  | 51.9     | 53.7           | 55.4            | 56.7                      | 58.2                      |
> | **GSM8K (%)** | 84.5     | 86.2           | 87.2            | 86.8                      | 89.7                      |
>
> | **Model**     | DeepSeek | DeepSeek + SFT | DeepSeek + HSFT | DeepSeek-HSFT + Reflexion | DeepSeek-HSFT + Cross-DPO |
> | ------------- | :------: | :------: | :------: | :------: |:------:  |
> | **MATH (%)**  | 46.8     | 49.2           | 50.9            | 51.2                      | 54.6                      |
> | **GSM8K (%)** | 82.9     | 84.5           | 85.7            | 85.8                      | 88.2                      |
>
> **Q4: Generalize SuperCorrect to More Tasks.**
>
> **A4:**
> 1. We sincerely thank you for your acknowledgement and insightful suggestions for our method. Actually, we also plan to evaluate our method on various tasks and benchmarks. To generalize to a wider range of tasks,  we need to curate more high-quality dataset, including hierarchical thought based reasoning dataset and correction trace dataset. This process is time-consuming and requires a lot of resources. Unfortunately, due to these limitations, we are currently only able to provide evaluations on Math tasks.
>
> 2. We are currently working on curating high-quality datasets for different tasks, and we have applied our method to a wider range of tasks. The results are promising. Thank you again for your kind suggestion, and please stay tuned for updates on our exploration of additional benchmarks.

---

> ### Comment · Reviewer_Bwuv · 2024-11-22
> **Remaining Concerns**
>
> I would like to thank the authors for their detailed feedback.
>
> **Regarding Q1-A1 (i.e cross-model DPO):** I think these results indeed look promising and I would like to see them in the supplement for the final version. This gives a clear quantitative sense on how far your method brings the student model and the teacher model closer to each other, and can serve as a template benchmark for follow up works. Please also refer to this section of the supplement within the main paper.
>
> **Regarding Q2-A2 (i.e HSFT):** While I appreciate the extra prompts tested and the reported results, I think my concern still remains here. Basically, the claim is that the "Hierarchical" prompt is being beneficial for improving the reasoning. (correct me if I'm wrong on this). Yet, your prompt severely differs from the baseline prompt (i.e. CoT) in many ways. This makes is unclear if the gains are coming from the formatting, or from the hierarchical aspect. I think the non-XML format is good to be tested, but still, it's severely different from the CoT prompt.
>
> For example, a *comparable* non-hierarchical CoT prompt to be served as a baseline would be something in lines of the following:
>
> ------------
> **Task:** Solve the following math problem step-by-step. Each step should be clearly numbered (e.g., Step 1, Step 2).
>
> 1. **Identify the Problem:** Begin by stating the problem clearly.
>
> 2. **Step-by-Step Breakdown:** For each step:
>         Clearly describe the action you are taking. Assess whether this step is challenging or tricky. If it is, provide a detailed explanation and analysis to help clarify your reasoning process.
>
> 3. **Combine Results:** After identifying all relevant factors or multiples, compile the results carefully, ensuring no duplicates are counted.
>
> 4. **Count Unique Results:** Count the total number of unique results identified from the previous steps.
>
> 5. **Final Answer:** Present the final answer clearly at the end.
> ------------
>
> Notice how this is exactly the same prompt, with the same exact wording, except for the missing **Generalization** step thus not having the hierarchical aspect. From what I understand from current version of the paper is claiming (Lines 193&194), it seems to suggest that the main contribution is the *Generalization* step of the thought. If this is not the case, and you are claiming the entire prompt style as your contribution, it should be stated more clearly. Note that the above prompt was just an example to show what I mean by an *comparable CoT baseline*.

---

> ### Comment · Reviewer_Bwuv · 2024-11-22
>
> Thanks for the detailed results for Q3 (regarding missing rows in Table 2)
>
> I would say that this table should at least be in the supplement and be referred to in the caption of  Table 2.
>
> Thanks!

---

> ### Author Response · Authors · 2024-11-23
> **Response to Remaining Concerns**
>
> We sincerely thank you for your insightful suggestions for the update of our final version and evaluation of prompt style.  Here we would address your remaining concerns:
>
> - Q1-A1 and Q3-A3: We have followed your suggestions and add these new quantitative results to the supplement. Please **refer to Appendix.E in our updated maniscript** for detailed analysis. We also refer to this section of the supplement within the main paper (denoted in blue color). We believe that incorporating these experimental analyses will enhance the clarity and credibility of our paper. Thanks for your suggestions.
>
> - Q2-A2: First, we have to clarify that our main contribution in HSFT stage is that we proposed hierarchical thought template, which can also be considered as a hierarchical prompt style, instead of adding the generalization step. As mentioned in L186-196, the motivation of our hierarchical reasoning is inspired from BoT [1]. However, we found that high-level thought is insufficient for solving complex math problems, and we further design hierarchical reasoning which includes both generalization part for summarization and detailed reasoning thought for more specific reasoning.
>
> For more comprehensive analysis about the evaluation of prompt style. We follow your suggestions to conduct additional quantitative experiments to verify the effectiveness of our hierarchical reasoning prompt and explore the impact of generalization step. Here we curated two different dataset  based on the same 100k math problems in the previous  experiment, and trained Qwen2.5-Math-Instruct, Llama3.1-8B-Instruct and DeepSeek-Math-7B on these dataset with the same training settings. The results are added in the table below. From the table. We can observe that the hierarchical reasoning style significantly improves the model's overall performance. Although adding generalization steps helps the model better summarize tasks thereby further enhancing its performance, our experimental results indicate that the **hierarchical reasoning style we designed is the primary contribution** to the performance improvements in the HSFT stage. We have also updated our analysis and experiments **in the Appendix.E our updated manuscript**.
>
>
>
> | Models/Prompt Style      | CoT  | CoT + Hierarchical  (No Generalization) | CoT + Hierarchical  (With Generalization) | Hierarchical Prompt （Not XML） | Hierarchical Prompt (XML) |
> | ----- | :----: | :----:| :---: |:---: | :---:|
> | Qwen2.5-Math-7B-Instruct | 57.4 | 59.7  | 60.8  | 61.8 | 62.4   |
> | Llama3.1-8B-Instruct     | 52.6 | 53.3    | 53.6   | 53.7   | 54.1    |
> | DeepSeek-Math-7B         | 46.8 | 49.6    | 50.2   | 50.6     | 51.6    |
>
> *Here we present more prompts we used in the experiments for better demonstration*:
>
> ```CoT + Hierarchical  (No Generalization):```
>
> **Task:** Solve the following math problem step-by-step. Each step should be clearly numbered (e.g., Step 1, Step 2).
>
> 1. **Identify the Problem:** Begin by stating the problem clearly.
>
> 2. **Step-by-Step Breakdown:**
>
>    For each step:
>
>    * Clearly describe the action you are taking.
>
>    * Assess whether this step is challenging or tricky. If it is, provide a detailed explanation and analysis to help clarify your reasoning process.
>
> 3. **Combine Results:** After identifying all relevant factors or multiples, compile the results carefully, ensuring no duplicates are counted.
>
> 4. **Count Unique Results:** Count the total number of unique results identified from the previous steps.
>
> 5. **Final Answer:** Present the final answer clearly at the end.
>
> ```CoT + Hierarchical  (With Generalization):```
>
> **Task:** Solve the following math problem step-by-step. Each step should be clearly numbered (e.g., Step 1, Step 2).
>
> 1. **Identify the Problem:** Begin by stating the problem clearly.
>
> 2. **Step-by-Step Breakdown:**
>
>    For each step:
>
>    - Clearly describe the action you are taking.
>    - Assess whether this step is challenging or tricky. If it is, provide a detailed explanation and analysis to help clarify your reasoning process.
>
> 3. **Combine Results:** After identifying all relevant factors or multiples, compile the results carefully, ensuring no duplicates are counted.
>
> 4. **Count Unique Results:** Count the total number of unique results identified from the previous steps.
>
> 5. **Generalization:** Summarize the problem-solving process and the key insights gained. Discuss how this approach could be applied to similar problems or any patterns that emerged during the solution.
>
> 6. **Final Answer:** Present the final answer clearly at the end.
>
> We hope our response helps you better understand our method and addresses your concerns. If you have any further questions, please feel free to reach out for further discussion.
>
> [1] Yang L, Yu Z, Zhang T, et al. Buffer of Thoughts: Thought-Augmented Reasoning with Large Language Models[J]. NeurIPS 2024.

---

### Official Review · Reviewer_qT8r · 2024-11-04

**Soundness:** 3
**Presentation:** 3
**Contribution:** 2
**Rating:** 8
**Confidence:** 4

**Summary:**

The paper proposes SuperCorrect, mainly focuses on addressing the issue of unability of detecting erroneous steps in their reasoning steps of small language models. SuperCorrect firstly prompts powerful LLMs to generate correct CoT reasoning steps and their detailed thought templates, then employs cross-model collaborative DPO to enhance the inner reasoning and self-correct abilities of small language models. Extensive experiments have demonstrate the effectiveness of proposed SuperCorrect framework.

**Strengths:**

1. The paper proposes hierarchical thought template that more effectively captures the underlying reasoning mechanisms of powerful LLMs compared to simply prompting them to generate reasoning steps.
2. Cross-model collaboration applies step-level correction and DPO, which can provide small language models more detailed supervise signals and than solution-level training, thus leading to more powerful reasoning capabilities.
3. Extensive experiments are convicing, demonstrating the effectiveness of SuperCorrect.
4. The paper is well-structured and well-written.

**Weaknesses:**

Teacher LLMs generated contents lack evaluation. For example, the logic flaws and errors found by teacher LLMs in student LLMs generated reasoning steps (Line 331). In Line 333, are the analysis $a_i$ and correction $c_i$ annorated by humans or generated by teacher LLMs? If it’s the latter case, what are the quality of these generated contents?

It’s seems like the cross-model collaborative DPO is nearly identical to the Step-DPO [1] paper, with the main difference being that in Step-DPO, $a_i$ and $c_i$ are sampled directly from the policy model, whereas in this paper, they are annotated/generated by humans/teacher LLMs. Can you explain the core differences? Or are these simply parallel works?

[1] Lai, X., Tian, Z., Chen, Y., Yang, S., Peng, X. and Jia, J., 2024. Step-dpo: Step-wise preference optimization for long-chain reasoning of llms. arXiv preprint arXiv:2406.18629.

**Questions:**

See Weaknesses.

---

> ### Author Response · Authors · 2024-11-20
> **Response to Reviewer qT8r - Part 1**
>
> *We thank Reviewer qT8r for the constructive review and valuable feedback. We are glad that the reviewer found our paper to be well-structured and well-written, and that our proposed framework is effective. Additionally, we are pleased that the reviewer recognized that SuperCorrect-7B has powerful reasoning capabilities and that our experimental results are convincing Please see below for our responses to your comments.*
>
> **Q1: Quality of Teacher LLM Generated Content.**
>
> The analysis $a_i$ and correction $c_i$ are generated by frontier teacher LLMs such as GPT-4o/o1-mini, as mentioned in our experiment parts. We ensure the data generation quality from these aspects:
>
> **1. Additional Iterative Evaluation to Ensure Quality**
> To address potential errors in the generated content, our method incorporates an evaluation process during the dataset curation phase. We utilize an inspector LLM, which verifies the accuracy of the correction trace by comparing it against the input problem and the ground-truth solution. If issues are detected, the problematic parts are sent back to the teacher LLMs for revision. This iterative checking process continues until no errors remain, with a maximum of three iterations allowed.
>
> We present both quantitative and qualitative results of our evaluation method. The table below compares the correctness of correction traces generated by three different teacher LLMs across three datasets. The application of the Inspector LLM significantly improves the quality of the final correction traces compared to direct generation. Notably, for LLMs with advanced capabilities that already produce high-quality outputs, it still shows clear improvements. These results demonstrate that the Inspector LLM markedly enhances the accuracy of correction traces, especially for datasets where initial performance was lower. **This iterative evaluation method effectively ensures the quality and reliability of the generated content.**
>
> | Model/Dataset                                           | GSM8K | MATH  | GaoKao |
> | ------------------------------------------------------- | ----- | ----- | ------ |
> | Teacher LLM (GPT-4o-mini)                               | 100%  | 92.4% | 89.6%  |
> | Teacher LLM (GPT-4o-mini)  + Inspector LLM (o1-preview) | 100%  | **98.8%** | **96.2%**  |
> | Teacher LLM (GPT-4o)                                    | 100%  | 94.4% | 91.3%  |
> | Teacher LLM (GPT-4o)  + Inspector LLM (o1-preview)      | 100%  | **99.2%** | **97.5%**  |
> | Teacher LLM (o1-mini)                                   | 100%  | 98.2% | 94.8%  |
> | Teacher LLM (o1-mini)  + Inspector LLM (o1-preview)     | 100%  | **99.6%** | **98.7%**  |
>
> **2. Ensuring the Quality of Direct Generation**
>
> Above experiment results without Inspector LLM reveal that our direct generated correction traces have already been of high quality. We attribute this high quality to the following design:
> * **Leveraging Frontier Teacher LLMs:** To ensure the quality of content generated by the teacher LLM, we leverage state-of-the-art models, such as o1-mini, which are capable of identifying logical flaws and errors, and generating high-quality analysis and corrections, as shown in the quantitative results.
> * **Grounding Correction Traces with Ground-Truth Context:** To ensure the accuracy of the correction trace generated by the teacher LLM, as demonstrated in Appendix A, the prompt for generating analysis ($a_i$) and correction ($c_i$) is based on the input question along with the ground-truth solution. This approach grounds the correction trace with the ground-truth solution as context, thereby guaranteeing the accuracy of the generated content.

---

> ### Author Response · Authors · 2024-11-20
> **Response to Reviewer qT8r - Part 2**
>
> **Q2: Core Difference between Cross-model DPO and Step-DPO.**
>
> A2:
> 1. The core difference between our Cross-model DPO and Step-DPO is that Step-DPO only focuses on learning to **correct reasoning errors**,  whereas our Cross-model DPO focuses on learning to **correct self-correction errors**, which is a more high-level objective. More specifically, Step-DPO only learns from the **correct solution** without explicitly learning from the self-correction of erroneous steps or analyzing the causes of these errors. Thus, it primarily improves the reasoning ability of LLMs. In contrast to Step-DPO, our Cross-model DPO explictly learns from the **correct correction traces containing both error analysis $a_i$ and corrections $c_i$** for incorrect reasoning steps. This allows the model not only to learn the correct solution but also to gain a deeper understanding of the underlying causes of errors. As a result, our Cross-model DPO not only improves the reasoning ability of LLMs but also **enables LLMs to accurately locate and conduct self-correction on erroneous steps**.
>
> 2. Additionally, the choice of policy model (teacher model in Step-DPO) significantly affects the method's potential upper limits.
> Step-DPO utilizes a policy model that is the same as target model to stabilize the optimization process under an in-distribution setting.
> Our method, Cross-model DPO, utilizes a more powerful teacher LLM, and **enables the student LLM to learn content beyond its initial distribution towards better teacher LLMs, enabling it to solve previously unsolvable problems.** Thus, comparing with Step-DPO, our approach not only enhances model stability but also extends its potential capabilities. We additionally conduct a detailed comparative analysis in **the Appendix.C of our updated manuscript**.
>
>
> 3. To have more comprehensive understanding of the difference between Step-DPO with our Cross-model DPO, we conduct the same experiments with Step-DPO as mentioned in Fig 4, and we show the quantitative results in the table below. To ensure a fair comparison, we omitted the initial HSFT phase and applied Cross-model DPO directly to the base model which is the same as Step-DPO. As shown in the table, compared to Step-DPO, our method not only achieves **higher improvements across all topics**, we also observe that although Step-DPO shows consistent improvement across all topics, these improvements **did not vary significantly across different topics**. In contrast, our method shows a more significant improvement on the topics that are originally difficult for LLMs, further validating our claim that our method could help student LLM to solve previously unsolvable problems.
>
> | Model/Topic               | Precalculus | Prealgebra | Number Theory | Intermediate Algebra | Geometry | Counting & Probability | Algebra |
> | ------------------------------ | ----------- | ---------- | ------------- | -------------------- | -------- | ---------------------- | ------- |
> | Qwen2.5-Math                   | 0.3498      | 0.7635     | 0.4981        | 0.3433               | 0.4384   | 0.4979                 | 0.7355  |
> | Qwen2.5-Math + Step-DPO        | 0.3924 (+0.0426)  | 0.7863 (+0.0228) | 0.5237 (+0.0256) | 0.3914 (+0.0481) | 0.4671 (+0.0287) | 0.5369 (+0.0390) | 0.7664 (+0.0309)  |
> | Qwen2.5-Math + Cross-model DPO | 0.5012 (+0.1514)  | 0.7924 (+0.0289) | 0.6203 (+0.1222) | 0.4972 (+0.1539) | 0.4825 (+0.0441) | 0.5764 (+0.0785) | 0.7842 (+0.0487)  |

---

> > ### Comment · Reviewer_qT8r · 2024-11-20
> > **Official Comment**
> >
> > The authors have addressed all my concerns and I have raised my score.

---

> > > ### Author Response · Authors · 2024-11-20
> > > **Thanks for raising score**
> > >
> > > Dear Reviewer qT8r，
> > >
> > > Many thanks for raising score! We sincerely appreciate your valuable comments and your precious time in reviewing our paper!
> > >
> > > Warm Regards,
> > >
> > > The Authors

---

### Author Response · Authors · 2024-11-20
**Global Response**

We sincerely thank all the reviewers for their thorough reviews and valuable feedback. We are glad to hear that  our proposed framework is novel and generalizable (reviewer Bwuv and gRHY), the paper is well-written and presented (reviewer qT8r,Bwuv and hUwP), and the performance improvements demonstrated in experiments are significant and promising (all reviewers).

Here, we want to highlight the main contributions and novelties of our proposed framework, "SuperCorrect" :

**Hierarchical Thought Based Reasoning:** SupperCorrect introduces a novel first-stage finetuning paradigm that leverages hierarchical thought based reasoning to improve reasoning ability of LLMs. This approach not only significantly enhancing accuracy, but also enable LLMs to consistently generate reasoning thought during the reasoning process.

**Cross-model Collaborative DPO:**  SuperCorrect proposes a novel second-stage Cross-model Collaborative DPO which utilizes teacher LLM to correct the erroneous thought exposed in the self-correction traces of student LLMs. Regarding cross-model DPO innovatively learns to correct self-correction errors instead of only correcting reasoning errors, the student LLMs can focus on both correct and erroneous correction traces which help to better improve the self-correction ability and further break the original thought bottleneck.

**Empirical Validation:** Extensive experiments on MATH and GSM8K datasets demonstrates significant performance improvements over state-of-the-art models, including substantial accuracy improves and promising generalization ability to larger sized models and more various domains.

We summarize our responses to the reviewers' comments as follows:

* We additionally provide more examples and conduct more experiments to show the quality of our dataset and **updated our manuscript in Appendix.D**. (Reviewer qT8r and hUwp)

* We additionally conduct ablation experiments and provide more quantitative and qualitative comparisons for the difference between Step-DPO and our Cross-model DPO and **updated our manuscript in Appendix.C** (Reviewer qT8r)

* We additionally conduct ablation study on the impact of prompt style, and prsent quantitative analysis. (Reviewer Bwuv)

We reply to each reviewer's questions in detail below their reviews. Please kindly check out them. Thank you and please feel free to ask any further questions.

---

### Meta-Review · Area_Chair_3QJU · 2024-12-16

**Metareview:**

We recommend the paper to be accepted for Poster.

Below a more detailed description of the paper.

The paper introduces SuperCorrect, a two-stage framework to improve the mathematical reasoning of smaller models like Llama-3-8B and DeepSeekMath-Base. The papers shows that new method achieves SOTA performance across all models considered.
The main strengths (S#) of the paper are

- (S1)	the proposed methodology that seems to more effectively capture the reasoning mechanisms of powerful LLMs.
- (S2)	the proposed methodology, especially the cross-model DPO, can be applied to other domains.
- (S3)	the promising quantitative results obtained across different models, that achieve SOTA performance.
- (S4)	the paper is well written and easy to follow, also for people not familiar with the specific topic.

Many of the weaknesses pointed out by the reviewers were addressed by the authors.
Some weaknesses (W#) that may remain are the following:

- (W1)	The paper primarily focuses on small models and it is not clear whether the methodology proposed can scale to larger models. This in turn means that the methodology may be only applicable to a niche of available models; aspect that may still be useful, but limit its interest to the wider community.
- (W2)	The success of the proposed methodology relies on the quality of the fine-tuning datasets and the paired correction traces. The paper mentions constructing high-quality datasets, but it may face challenges in scenarios where such curated datasets are not available or the domain of interest is very niche.

Some discussions on how to overcome these weaknesses have been mentioned by the authors, but are not conclusively addressed, and may limit the success and adoption of the methodology.

**Additional Comments On Reviewer Discussion:**

The authors have been proactive in addressing the comments raised by the reviewers, and the reviewers were well engaged responding to the authors.

We agree with the reviewers comments, and recommendations, noting some of the weaknesses that we believe may remain and mentioned in the metareview.

No ethics review raised by the reviewers, and we agree with them.

---

### Decision · Program_Chairs · 2025-01-22

Accept (Poster)